# Continuous transport of a small fraction of plasma membrane cholesterol to endoplasmic reticulum regulates total cellular cholesterol

Rodney Elwood Infante[1,2]*, Arun Radhakrishnan[1]*

[1]Departments of Molecular Genetics, University of Texas Southwestern Medical Center, Dallas, United States; [2]Internal Medicine, University of Texas Southwestern Medical Center, Dallas, United States

**Abstract** Cells employ regulated transport mechanisms to ensure that their plasma membranes (PMs) are optimally supplied with cholesterol derived from uptake of low-density lipoproteins (LDL) and synthesis. To date, all inhibitors of cholesterol transport block steps in lysosomes, limiting our understanding of post-lysosomal transport steps. Here, we establish the cholesterol-binding domain 4 of anthrolysin O (ALOD4) as a reversible inhibitor of cholesterol transport from PM to endoplasmic reticulum (ER). Using ALOD4, we: (1) deplete ER cholesterol without altering PM or overall cellular cholesterol levels; (2) demonstrate that LDL-derived cholesterol travels from lysosomes first to PM to meet cholesterol needs, and subsequently from PM to regulatory domains of ER to suppress activation of SREBPs, halting cholesterol uptake and synthesis; and (3) determine that continuous PM-to-ER cholesterol transport allows ER to constantly monitor PM cholesterol levels, and respond rapidly to small declines in cellular cholesterol by activating SREBPs, increasing cholesterol uptake and synthesis.

*For correspondence: rodney.
infante@utsouthwestern.edu (REI);
arun.radhakrishnan@
utsouthwestern.edu (AR)

**Competing interests:** The authors declare that no competing interests exist.

## Introduction

Animal cells carefully control both their overall content of cholesterol as well as its distribution among organelles. Although present in the membrane of every organelle, 60–90% of a cell's cholesterol is concentrated in its plasma membrane (PM) (*De Duve, 1971*; *Lange et al., 1989*). In the PM, the concentration of cholesterol is ~45 mole% of total lipids (*Das et al., 2013*). How does PM acquire and maintain these high levels of cholesterol? Much has been learned about the two sources through which cells acquire cholesterol – (i) receptor-mediated endocytosis of cholesterol-rich low-density lipoprotein (LDL) (*Brown and Goldstein, 1986*), and (ii) biosynthesis (*Brown and Goldstein, 1990*). However, neither of these two sources is located in the PM. LDL-derived cholesterol from lysosomes and synthesized cholesterol from endoplasmic reticulum (ER) must be moved to PM by transport mechanisms that are not understood. Moreover, the SREBP regulatory network that controls rates of LDL uptake and cholesterol biosynthesis is located not in the PM, but in the ER (*Brown and Goldstein, 2009*). Thus, PM cholesterol levels must be sensed and this information must be transmitted to ER to ensure optimal cholesterol levels in PM. These sensing mechanisms are also not fully understood.

Recent studies have provided clues regarding how the organization of PM cholesterol into three distinct pools may play a role in regulating cholesterol levels (*Das et al., 2014*). These studies showed that one pool of PM cholesterol, ~15 mole% of PM lipids, is sequestered by sphingomyelin. A second pool, ~12 mole% of PM lipids, is sequestered by other membrane factors. Cholesterol in

**eLife digest** Cells are surrounded by a plasma membrane made mostly from oily molecules known as lipids. One of these lipids, called cholesterol, is essential for keeping this membrane stable. Cholesterol is partly produced within the cells at a specialized structure called the endoplasmic reticulum, and partly imported from the blood surrounding the cell. In the blood, cholesterol is shielded inside particles called low-density lipoprotein (or LDL for short), which is taken into the cell and then sent to another structure called the lysosome.

Inside the cell, cholesterol that is freshly produced in the endoplasmic reticulum or freshly imported into the lysosome, must be moved to the plasma membrane, where most of the cholesterol is located. Cholesterol levels are regulated by a 'control machinery' of proteins located in the endoplasmic reticulum. To keep the cholesterol levels constant, the endoplasmic reticulum needs to be in continual communication with the plasma membrane. However, the mechanisms by which cholesterol is transported between membranes are still poorly understood. Here, Infante and Radhakrishnan report a new tool to study how cholesterol is transported in human and hamster cells. The tool, which is based on part of a bacterial protein, traps cholesterol in the plasma membrane and prevents it from moving to the endoplasmic reticulum, and thus from updating the control machinery about cholesterol levels.

From this inhibition, it is inferred that a stream of cholesterol constantly travels from the plasma membrane back to endoplasmic reticulum. This way, proteins in the endoplasmic reticulum can monitor the cholesterol levels in the plasma membrane in real-time. The endoplasmic reticulum responded rapidly even to small declines in cholesterol levels by activating genes that increase cholesterol production or the amount of cholesterol imported via the LDL pathway. Further work showed that cholesterol derived from LDL travels from the lysosome directly to the plasma membrane to maintain optimal cholesterol levels. It then moves to the endoplasmic reticulum to signal that cholesterol levels in the cell have been satisfied.

The findings and tools described in this study will help to further investigate the mechanisms underlying the transport of cholesterol between the different membranes and structures in a cell. A next step will be to see if the mechanisms that apply to distribution of imported cholesterol from lysosomes, also apply to the cholesterol produced in the endoplasmic reticulum.

excess of these two sequestered pools comprises a third pool that signals cholesterol excess to the regulatory machinery in the ER, thereby avoiding cholesterol overaccumulation while ensuring optimal cholesterol levels in PM. Defining these pools of cholesterol in PM was made possible by the use of mutant versions of Perfringolysin O (PFO), a bacterial toxin that binds specifically to accessible cholesterol in membranes (*Das et al., 2013*; *Flanagan et al., 2009*; *Sokolov and Radhakrishnan, 2010*). Unfortunately, pore formation at 37°C by the PFO-derived probes prevented us from studying PMs of living cells, and restricted their use to PMs of cells that had been chilled to 4°C (*Das et al., 2014*, *2013*).

We overcame the undesired lytic properties of PFO probes by taking advantage of recent work where we showed that sub-domains of PFO and anthrolysin O (ALO), another closely related toxin, bound membrane cholesterol but did not form large oligomeric pores in red blood cells at 37°C (*Gay et al., 2015*). In the studies described here, we used the cholesterol-binding domain 4 of ALO, hereafter referred to as ALOD4, due to the protein's higher stability. When non-lytic ALOD4 was used to study cells at 37°C, we made the surprising observation that ALOD4 was not just a reporter of PM cholesterol accessibility; it also trapped cholesterol at PM and specifically inhibited PM-to-ER cholesterol transport. Using this new cholesterol trafficking inhibitor, we traced the route taken by LDL-derived cholesterol to show that it travels directly from lysosomes to PM. We also show that trapping as little as 1% of PM cholesterol triggers the activation of the SREBP regulatory network in ER. This activation leads to increased cholesterol production and uptake, thus enabling a rapid and switch-like response to restore PM cholesterol to optimal levels.

## Results

### Recombinant ALOD4 does not lyse CHO-K1 cells at 37°C

To develop a tool to study PM cholesterol in cells growing at 37°C, we overexpressed His$_6$-tagged ALOD4 in bacteria, and purified the resulting recombinant protein as described in Materials and methods. Gel-filtration chromatography of purified ALOD4 showed that it eluted as a single sharp peak (*Figure 1A*, *blue*), and its homogeneity was confirmed by Coomassie staining (*Figure 1A inset*). We also produced fluorescently-labeled versions of ALOD4 as described previously (*Gay et al., 2015*). Gel-filtration chromatography and Coomassie staining combined with fluorescence gel imaging of ALOD4 labeled with Alexa Fluor 488 (fALOD4-488) or Alexa Fluor 647 (fALOD4-647) showed that their elution profiles and homogeneity were similar to those of unlabeled ALOD4 (*Figure 1A*).

We next tested whether ALOD4 would form pores in CHO-K1 cells at 37°C. As a positive control for pore formation, we purified the full-length version of ALO (ALOFL) that forms large oligomeric pores in cells (*Bourdeau et al., 2009*; *Gay et al., 2015*). When added to CHO-K1 cells, ALOFL permeabilized the PM as revealed by immunoblotting of the medium for two cytosolic proteins, lactate dehydrogenase (LDH) and ubiquitin-activating enzyme (E1) (*Figure 1B*, *lanes 2–4*). Similar release of cytosolic content was observed when the cells were lysed with SDS (*Figure 1B*, *lane 9*); however, no such effect was observed when cells were treated with ALOD4, even at the highest concentration of 10 μM (*Figure 1B*, *lanes 6–8*). This result is consistent with earlier studies where addition of the non-lytic domain 4 of PFO or ALO to the external medium resulted in labeling of the PM of intact cells, but not of internal organelle membranes (*He et al., 2017*; *Ishitsuka et al., 2011*; *Maekawa et al., 2016*; *Shimada et al., 2002*).

### Binding of ALOD4 to PMs triggers activation of SREBPs

Although we initially designed ALOD4 as a live-cell sensor of accessible cholesterol levels in PM, the question arose as to whether the binding of ALOD4 to PM cholesterol would perturb intracellular

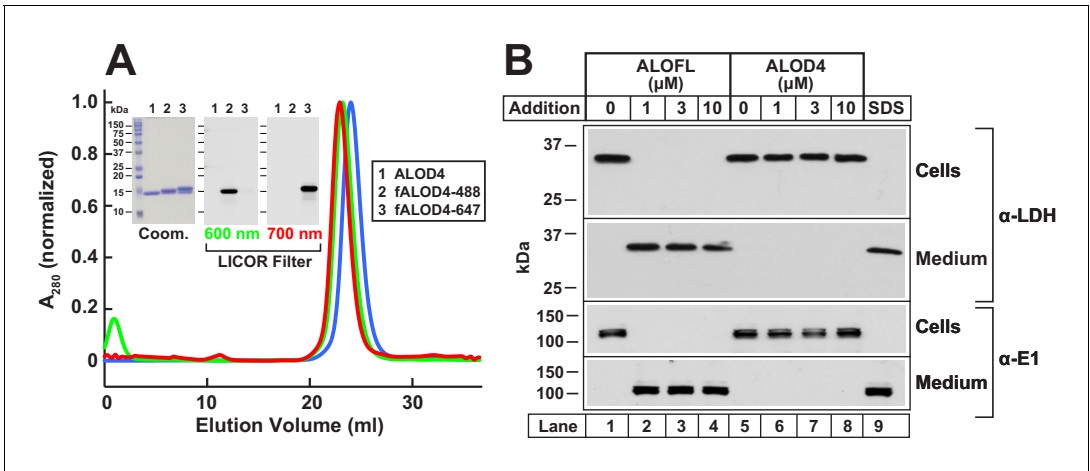

**Figure 1.** Biochemical characterization and non-lytic properties of ALOD4. (**A**) Gel filtration chromatography of purified proteins. Recombinant ALOD4 was purified and labeled with Alexa Fluor 488 (fALOD4-488) or Alexa Fluor 647 (fALOD4-647) fluorescent dyes as described in Materials and methods. Buffer B (1 ml) containing 0.8 mg of ALOD4, fALOD4-488, or fALOD4-647 was loaded onto a Tricorn 10/300 Superdex 200 column and chromatographed at a flow rate of 0.5 ml/min. Absorbance at 280 nm (A$_{280}$) was monitored continuously to identify ALOD4(blue), fALOD4-488(green), or fALOD4-647(red) proteins. Maximum A$_{280}$ values for each protein (ALOD4: 390 mAU, fALOD4-488: 231 mAU, and fALOD4-647: 279 mAU) are normalized to one. (*Inset*) 3 μg of each protein was subjected to 15% SDS/PAGE and stained with Coomassie (left) or imaged with the 600 nm filter (middle) or the 700 nm filter (right) on a LICOR instrument. (**B**) Release of cytosolic proteins from CHO-K1 cells into media after incubation with ALOFL, but not ALOD4. On day 0, CHO-K1 cells were set up in medium B at a density of 6 × 10$^4$ cells/well of 48-well plates. On day 1, media was removed, and cells were washed with 500 μl of PBS followed by addition of either 200 μl of medium C with the indicated concentration of ALOFL or ALOD4, or with 200 μl of buffer C containing 1% SDS detergent. After incubation for 1 hr at 37°C, media was collected and cells were harvested as described in Materials and methods. Equal aliquots of cells and media (10% of total) were subjected to immunoblot analysis as described in Materials and methods. *Coom,* Coomassie.

cholesterol distribution. To answer this question, we incubated CHO-K1 cells with ALOD4 for 1 hr at 37°C, and then processed the cells for immunoblot analysis. Not surprisingly, a dose-dependent increase in binding of ALOD4 to cells was observed (*Figure 2A*, *top panel*). Most of the added ALOD4 was unbound and remained in the medium (*Figure 2A*, *two bottom panels*). As described later, we used fluorescently-labeled ALOD4 to quantify the kinetics and stoichiometry of ALOD4 binding to PMs (see Figure 4).

To examine the consequence of ALOD4 binding to the PMs of these cells, we conducted immunoblot analysis of SREBP1 and SREBP2, transcription factors that respond to declines in cellular cholesterol by activating genes encoding cholesterol biosynthetic enzymes and the LDL receptor that mediates uptake of cholesterol-rich LDL (*Horton et al., 2003*). After being synthesized in the ER, both SREBPs bind to Scap, a cholesterol-sensing membrane protein that escorts SREBPs from ER to Golgi when ER cholesterol levels are below a threshold level of ~5 mole% of total ER lipids (*Brown and Goldstein, 2009*). In the Golgi, Site-1 protease and Site-2 protease sequentially cleave SREBPs, generating an active transcription factor fragment that travels to the nucleus to upregulate lipogenic genes, eventually raising cholesterol levels in cells and in ER. When ER cholesterol rises above the threshold concentration of ~5 mole% of total ER lipids, cholesterol binds to Scap and promotes Scap's binding to Insigs, ER retention proteins. These interactions cause a conformational change in Scap, preventing its transport from ER to Golgi. Transport of SREBPs to Golgi is also blocked, and thus the proteolytic activation of SREBPs does not occur. As a result, cellular cholesterol levels decline and return to optimal levels. Activation of SREBPs is thus finely tuned to cellular cholesterol levels (*Brown and Goldstein, 2009*; *Goldstein and Brown, 2015*).

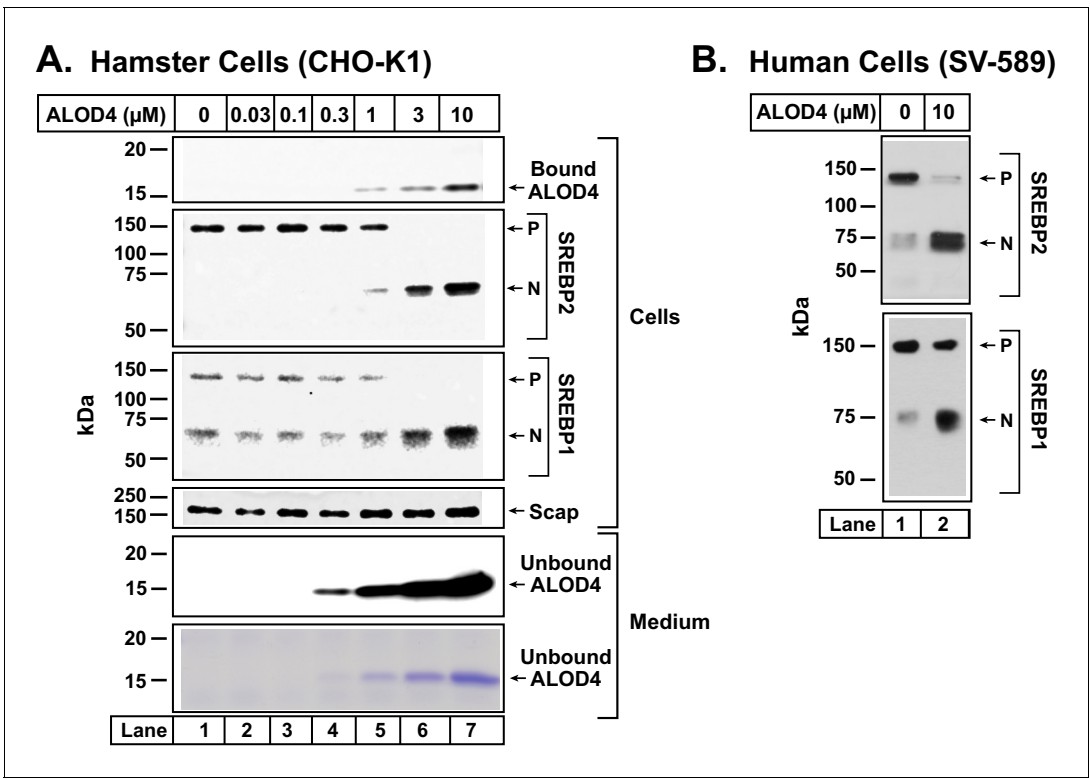

**Figure 2.** ALOD4 triggers activation of SREBP transcription factors in hamster and human cells. (A–B) On day 0, CHO-K1 cells were set up in medium B at a density of $6 \times 10^4$ cells/well of 48-well plates (A) and SV-589 cells were set up in medium H at a density of $4 \times 10^4$ cells/well of 48-well plates (B). On day 1 (A) or day 2 (B), media was removed, cells were washed with 500 µl of PBS followed by addition of 200 µl of medium C (A) or medium G (B) with the indicated concentrations of ALOD4. After incubation for 1 hr at 37°C, media was collected, and cells were harvested as described in Materials and methods. Equal aliquots of cells and media (10% of total) were subjected to immunoblot analysis or Coomassie staining as described in Materials and methods. P = precursor form of SREBP1 or SREBP2; N = cleaved nuclear form of SREBP1 or SREBP2.

As cells growing in lipoprotein-rich FCS were well supplied with cholesterol, almost all of their SREBP2 and about half of their SREBP1 were in their precursor ER forms (*Figure 2A*, *second and third panels, lane 1*). As increasing amounts of ALOD4 bound to these cells, we detected increasing amounts of the proteolyzed nuclear forms of both SREBP2 and SREBP1, and a corresponding decline in their precursor forms (*Figure 2A*, *second and third panels, lanes 1–7*). Levels of Scap were not affected by ALOD4 binding (*Figure 2A*, *fourth panel, lanes 1–7*). ALOD4 also triggered the activation of SREBP1 and SREBP2 in human fibroblast cells (SV-589) (*Figure 2B*).

To test whether ALOD4's triggering of SREBP2 activation required its binding to PM cholesterol, we compared the effect of ALOD4 to that of a purified mutant version that can no longer bind membrane cholesterol (*Gay et al., 2015*), designated as ALOD4(Mut). As shown in *Figure 3A*, when incubated with CHO-K1 cells, ALOD4 bound to cells (*top panel, lanes 1–4*) and triggered activation of SREBP2 (*middle panel, lanes 1–4*), whereas ALOD4(Mut) did not bind to cells (*top panel, lanes 5–8*) or trigger activation of SREBP2 (*middle panel, lanes 5–8*). Coomassie staining of the medium confirmed that similar amounts of ALOD4 and ALOD4(Mut) were added to cells (*Figure 3A*, *bottom panel, lanes 1–8*).

The results of *Figures 2* and *3A* were reminiscent of previous studies where SREBP activation was triggered by depleting cells of sterols, either by incubation in lipoprotein-poor serum (*Wang et al., 1994*) or by cholesterol extraction from PMs by cyclodextrin reagents (*Yang et al., 2002*). If ALOD4 blocked receptor-mediated endocytosis of lipoproteins, then the net result would be the same as incubation of cells in lipoprotein-poor serum. To test this possibility, we incubated CHO-K1 cells with ALOD4 in lipoprotein-rich FCS as well as in lipoprotein-poor serum (LPDS). As shown in *Figure 3B*, we observed similar binding of ALOD4 to cells (*top panel*) and similar triggering of SREBP2 activation (*bottom panel*) both in FCS (*lanes 1–5*) and LPDS (*lanes 6–10*) incubation conditions. See Figure 6 later for a more detailed analysis of LDL uptake and degradation by sterol-depleted cells in the presence of ALOD4.

We then tested whether ALOD4 triggered SREBP activation by removing cholesterol from cells. For this experiment, we compared ALOD4's effects to that of HPCD, a commonly used cyclodextrin that extracts cholesterol from cells and lowers the overall cellular cholesterol content (*Ohtani et al., 1989*; *Radhakrishnan et al., 2008*). As shown in *Figure 3C*, incubation of CHO-K1 cells with increasing concentrations of HPCD or ALOD4 both triggered similar activation of SREBP2 (compare *lanes 1–3* to *lanes 4–6*). The HPCD-induced increase in SREBP2 processing was accompanied by a decrease in total cellular cholesterol from ~30 µg/mg protein to ~18 µg/mg protein (*Figure 3C*, *lanes 1–3*). However, the ALOD4-induced increase in SREBP2 processing was observed even though there was no significant change in total cellular cholesterol (*Figure 3C*, *lanes 4–6*). Together, the experiments of *Figure 3* suggest that ALOD4 triggers SREBP2 activation by binding to PM cholesterol and preventing its transport to ER, and not by blocking lipoprotein uptake or depleting cellular cholesterol. To determine whether ALOD4-induced activation of SREBPs led to upregulation of target genes in a manner similar to that induced by cholesterol depletion, we isolated total RNA from cells treated with ALOD4 or HPCD and measured the levels of mRNAs for HMG CoA reductase and LDL receptor by quantitative real-time PCR. As shown in *Figure 3D*, expression of mRNAs for both these SREBP target genes was increased by ALOD4 treatment, and these increases were similar to that observed after HPCD treatment. No change was observed in mRNA levels of actin, which is not a SREBP target gene.

## Kinetics of binding of ALOD4 to PMs

To further understand ALOD4's effect on PM-to-ER cholesterol transport, we measured the kinetic parameters of this process. For more accurate quantification of binding of ALOD4 to PMs, we supplemented ALOD4 with tracer amounts (~5% of total protein) of fALOD4-488 or fALOD4-647, fluorescently-labeled versions of ALOD4 that trigger SREBP2 activation in CHO-K1 cells with the same concentration dependence as ALOD4 (*Figure 4A*, compare *lanes 1–4* to *lanes 5–8 and lanes 9–12*). We first measured the rate of association of ALOD4 to PMs of CHO-K1 cells. We detected rapid binding of ALOD4, both by immunoblot analysis (*Figure 4B*, *upper panel, lanes 1–7*) and by fluorescence measurements (*Figure 4C*, *black filled circles*). Half-maximal binding occurred at 10 min and saturation was reached after 40 min. Once ALOD4 bound to PMs, there was a lag time before triggering of SREBP2 activation at 60 min (*Figure 4B*, *lower panel, lanes 1–7*, and densitometry quantification in *Figure 4C*, *red filled circles*). As a control, ALOD4(Mut) showed no significant binding to

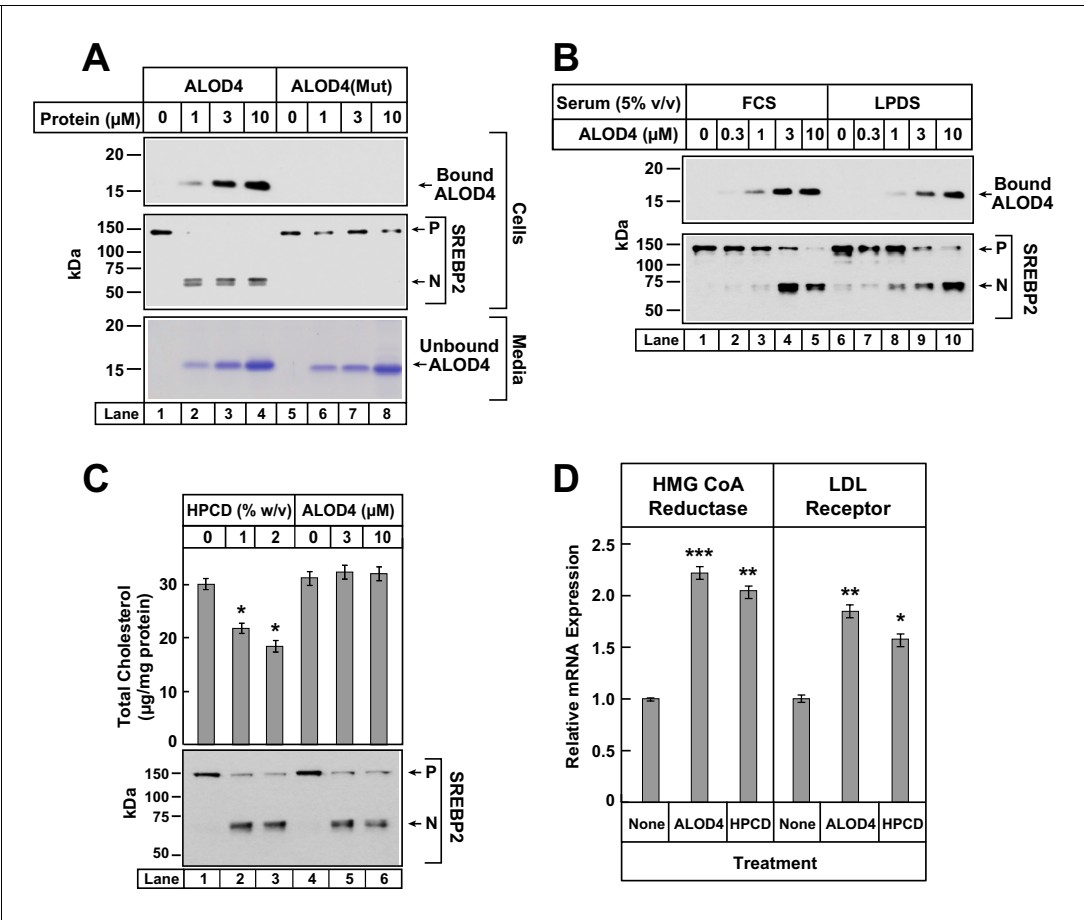

**Figure 3.** ALOD4 binding to cells activates SREBP2 transcription factors without changing cellular cholesterol levels. (A–B) Immunoblot analysis of CHO-K1 cells after incubation with ALOD4 or ALOD4(Mut) proteins in lipoprotein-rich or lipoprotein-poor serum. On day 0, CHO-K1 cells were set up in medium B at a density of either $3 \times 10^4$ cells/well of 48-well plates (A) or $6 \times 10^4$ cells/well of 48-well plates (B). On day 1 (B) or day 2 (A), media was removed, cells were washed with 500 µl of PBS and then the following additions were made: 200 µl of medium C with indicated concentrations of ALOD4 or ALOD4(Mut) (A) or 200 µl of lipoprotein-rich medium C or lipoprotein-poor medium F with indicated concentrations of ALOD4 (B). After incubation for 1 hr at 37°C, media was collected, and cells were harvested as described in Materials and methods. Equal aliquots of cells and media (10% of total) were subjected to immunoblot analysis or Coomassie staining as described in Materials and methods. (C) Cellular cholesterol levels decline after treatment with HPCD, but not after treatment with ALOD4. On day 0, CHO-K1 cells were set up in medium B at a density of $2.5 \times 10^5$ cells/60 mm dish. On day 3, media was removed, followed by addition of 200 µl of medium C with the indicated concentration of HPCD or ALOD4. After incubation for 1 hr at 37°C, media was removed and cells were harvested as described in Materials and methods. An aliquot of cells (5% of total) was used for immunoblot analysis (25 µg/lane), and the remainder was used for quantification of cholesterol as described in Materials and methods. Each column represents the mean of cholesterol measurements from three independent experiments, and error bars show the standard error (*top panel*). Asterisks denote level of statistical significance (Student *t* test) between cells treated without and with HPCD: *p<0.05. Immunoblot analysis of the cells from one of the three experiments is shown in the *bottom panel*. (D) ALOD4 treatment causes increases in mRNA levels of HMG CoA Reductase and LDL receptor genes. On day 0, CHO-K1 cells were set up in medium B at a density of $5 \times 10^5$ cells/100 mm dish. On day 2, media was removed, followed by addition of 2 ml of medium C in the absence or presence of ALOD4 (5 µM) or HPCD (1% w/v). After incubation for 4 hr at 37°C, media was removed, and cells were harvested for measurement of the indicated mRNA by quantitative RT-PCR as described in Materials and methods. For each gene, the amount of mRNA from untreated cells is set to 1, and mRNA amounts from cells treated with ALOD4 or HPCD are expressed relative to this reference value. Each column represents the mean of relative mRNA values measured in three independent experiments, and error bars show the standard error. Asterisks denote level of statistical significance (Student *t* test) between cells treated without and with ALOD4 or HPCD: *p<0.05; **p<0.01; ***p<0.001. The average $C_t$ values for actin (invariant control) were 15.38, 15.31, and 15.18 for the untreated, ALOD4-treated, and HPCD-treated conditions, respectively. The average $C_t$ values for HMG CoA Reductase and LDL receptor were 21.3 and 22.3, respectively, for the untreated condition. *P* = precursor form of SREBP2; *N* = cleaved nuclear form of SREBP2.

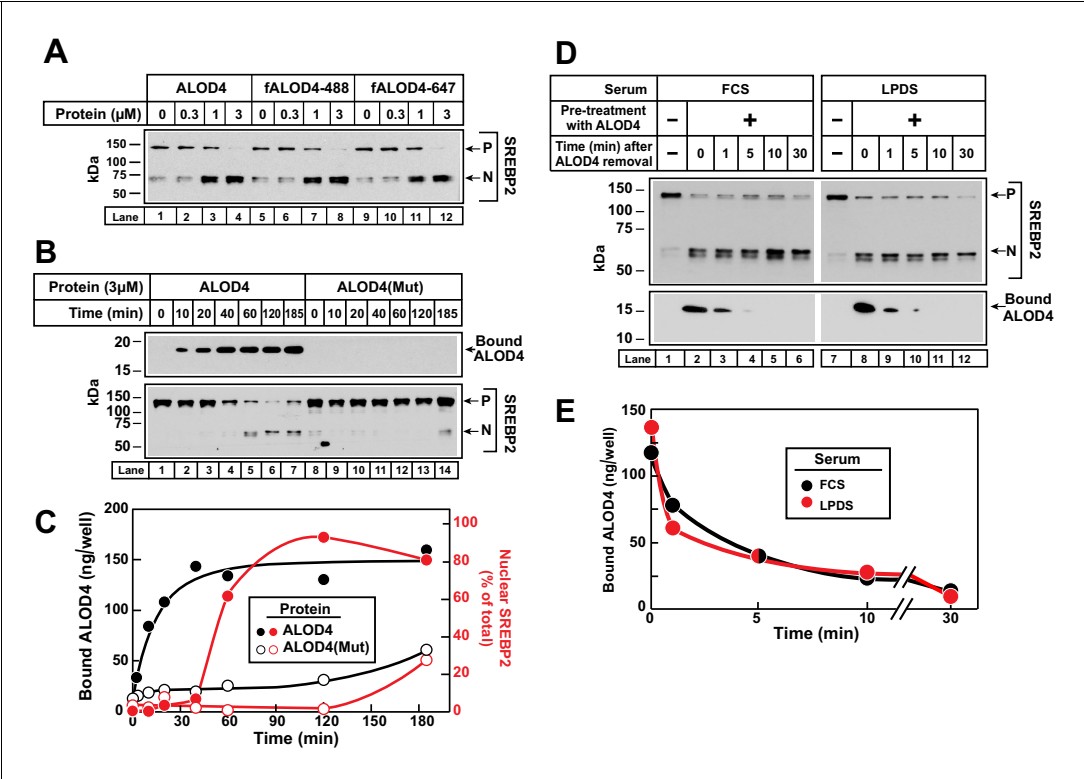

**Figure 4.** Kinetics of ALOD4 binding to CHO-K1 cells. (**A**) Fluorescence labeling of ALOD4 does not affect its ability to trigger SREBP2 activation. Recombinant ALOD4 was purified and labeled with Alexa Fluor 488 (fALOD4-488) or Alexa Fluor 647 (fALOD4-647) fluorescent dyes as described in Materials and methods. On day 0, CHO-K1 cells were set up in medium B at a density of $3 \times 10^4$ cells/well of 48-well plates. On day 2, media was removed, and cells were washed with 500 μl of PBS followed by addition of 200 μl of medium C with the indicated concentrations of ALOD4, fALOD4-488, or fALOD4-647. After incubation for 1 hr at 37°C, media was removed, and cells were harvested and subjected to immunoblot analysis as described in Materials and methods. (**B, C**) Association rate. On day 0, CHO-K1 cells were set up in medium B at a density of $6 \times 10^4$ cells/well of 48-well plates. On day 1, media was removed, and cells were washed with 500 μl of PBS followed by addition of 200 μl of medium C containing 3 μM of either ALOD4 or ALOD4(Mut). For quantification purposes, ~5% of ALOD4 or ALOD4(Mut) proteins were labeled with Alexa Fluor 488 dyes. After incubation for the indicated times at 37°C, the cells were harvested and subjected to either immunoblot analysis for detection of SREBP2 processing and bound ALOD4 (**B**) or fluorescence analysis of bound ALOD4 (**C**) as described in Materials and methods. LICOR quantification of SREBP2 from (**B**) is expressed as the amount of nuclear form relative to the total (precursor plus nuclear) (**C**, red). (**D, E**) Dissociation rate. On day 0, CHO-K1 cells were set up in medium B at a density of $6 \times 10^4$ cells/well of 48-well plates. On day 1, media was removed, and cells were washed with 500 μl of PBS followed by addition of 200 μl of lipoprotein-rich medium C or lipoprotein-poor medium F with 3 μM of ALOD4. For quantification purposes, ~5% of ALOD4 or ALOD4(Mut) proteins were labeled with Alexa Fluor 647 dyes. After incubation for 1 hr at 37°C, media was removed, and cells were washed twice with 500 μl of PBS followed by addition of 200 μl of medium C or medium F without ALOD4. After incubation for the indicated times at 37°C, the cells were harvested and subjected to either immunoblot analysis for detection of SREBP2 processing and bound ALOD4 (**D**) or fluorescence analysis of bound ALOD4 (**E**) as described in Materials and methods. Curves are drawn merely to guide the eye and do not represent a fit. *P* = precursor form of SREBP2; *N* = cleaved nuclear form of SREBP2.

PMs (*Figure 4B*, *upper panel, lanes 8–14,* and *Figure 4C*, *black open circles*), and triggered minor SREBP2 activation only at the very longest incubation periods used (*Figure 4B*, *lower panel, lanes 8–14,* and densitometry quantification in *Figure 4C*, *red open circles*).

Next, we measured the rate of dissociation of ALOD4 from PMs of CHO-K1 cells. We incubated cells with 3 μM ALOD4 for 1 hr, after which the ALOD4-containing medium was removed and replaced with medium without ALOD4. Consistent with our previous observations, ALOD4 triggered SREBP2 activation in both lipoprotein-rich FCS and lipoprotein-poor LPDS (*Figure 4D*, *upper panel*, compare *lane 1 to 2, and lane 7 to 8*). ALOD4 that had bound to cells during the 1 hr incubation was detected both by immunoblot analysis (*Figure 4D*, *lower panel, lanes 2 and 8*) and by fluorescence measurements (*Figure 4E*). Dissociation of this bound ALOD4 was rapid, with ~90% detaching from cells within 5 min (*Figure 4D*, lower panel, *lanes 2–6 and 8–12*, and *Figure 4E*). In addition to

providing insights into the nature of ALOD4's interaction with PM cholesterol, the rapid dissociation of bound ALOD4 from PMs also alleviated concerns that PM-bound ALOD4 may interfere with previously established methods to purify PM and ER membranes that we employ later for the analysis of *Figure 5* (*Das et al., 2013*; *Radhakrishnan et al., 2008*). One might expect that dissociation of bound ALOD4 would restore PM-to-ER cholesterol transport and suppress the activation of SREBP2 transcription factors. However, we did not observe any such reversal during the short 30 min duration of this experiment. This is not surprising since previous studies have shown that suppression of SREBP2 activation by cholesterol delivered in complexes with MCD or in lipoproteins requires 1–2 hr (*Radhakrishnan et al., 2008*). As shown later, suppression of SREBP2 activation after dissociation of bound ALOD4 was indeed observed after 4 hr (see Figure 7C).

## Binding of ALOD4 to PMs lowers ER cholesterol levels without changing whole cell or PM cholesterol levels

Our results so far suggest that the binding of ALOD4 to PMs blocks transport of PM cholesterol to ER, which leads to a decline in ER cholesterol levels even though overall cellular cholesterol levels do not change (*Figure 3C*). The decline in ER cholesterol level is inferred from the triggering of SREBP activation, which has been shown to occur when ER cholesterol levels drop below a threshold concentration of ~5 mole% (*Radhakrishnan et al., 2008*). We next sought to directly determine whether ALOD4 treatment lowers ER cholesterol levels. For this experiment, we set up three sets of CHO-K1 cells in lipoprotein-rich FCS. We then maintained one set of cells in FCS, and incubated the other two sets with either ALOD4 or cholesterol-depleting HPCD. After treatment for 1 hr, aliquots of cells

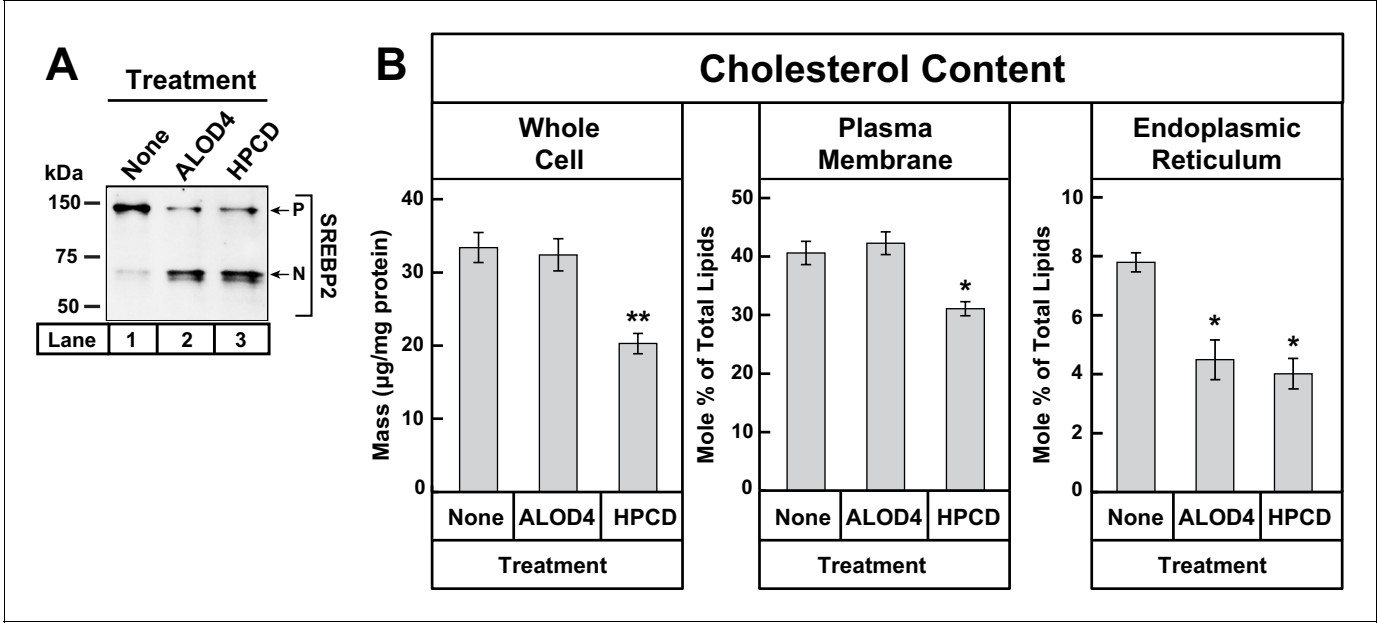

**Figure 5.** ALOD4 triggers activation of SREBP2 by lowering ER cholesterol levels while leaving PM cholesterol levels unchanged. (**A–B**) On day 0, CHO-K1 cells were set up in medium B at a density of $5 \times 10^5$ cells/100 mm dish (20 dishes/replicate/condition). On day 1, media was removed and fresh medium B was added. On day 2, media was removed, cells were washed with 5 mL PBS, followed by addition of 2 ml of medium B without or with ALOD4 (5 μM) or HPCD (1% w/v). After incubation for 1 hr at 37°C, media was removed, and cells were washed twice with 5 ml of PBS. After the washes, 5 ml of PBS was added to each dish and cells were harvested. An aliquot of cells (one dish/replicate/condition) was used for immunoblot analysis and quantification of whole cell cholesterol as described in Materials and methods. Another aliquot of cells (two dishes/replicate/condition) was used for PM purification and quantification of PM cholesterol, as described in Materials and methods. The remainder of cells (17 dishes/replicate/condition) was used for ER purification and quantification of ER cholesterol, as described in Materials and methods. Immunoblot of the cells from one of the three replicates for each treatment condition is shown in (**A**), and whole cell, PM, and ER cholesterol levels are shown in (**B**). Each column represents the mean of cholesterol measurements from triplicate assays, and error bars show the standard error. Asterisks denote level of statistical significance (Student *t* test) between cells treated without and with ALOD4 or HPCD: *p<0.05; **p<0.01. P = precursor form of SREBP2; N = cleaved nuclear form of SREBP2.

from each set were subjected to immunoblot analysis for SREBP2, whole cell cholesterol quantification, PM purification and quantification of PM cholesterol (*Das et al., 2013*), and ER purification and quantification of ER cholesterol (*Radhakrishnan et al., 2008*). Consistent with the results of *Figure 3C*, treatment of cells with ALOD4 or HPCD both triggered similar activation of SREBP2 (*Figure 5A*, lanes 2 and 3), however the treatments had distinct effects on cellular cholesterol distribution. Compared to the FCS-treated set, ALOD4 treatment did not significantly alter cholesterol levels in the whole cell or in PM, whereas HPCD treatment lowered total cellular cholesterol from ~33 µg/mg protein to ~20 µg/mg protein and PM cholesterol from 41 mole% of total PM lipids to ~31 mole% of total PM lipids (*Figure 5B*). A different relationship was found when we quantified ER cholesterol levels. The ER cholesterol content of FCS-treated cells was ~8 mole% of total ER lipids and this value declined by a similar degree to ~4 mole% of total ER lipids when treated with ALOD4 or HPCD (*Figure 5B*).

## LDL-derived cholesterol travels from lysosomes directly to PM

The above experiments demonstrate that ALOD4 blocks the transport of cholesterol from PM to ER. To determine if this property of ALOD4 would allow us to clarify the intracellular trafficking route of LDL-derived cholesterol, we first depleted CHO-K1 cells of cholesterol by incubation with HPCD, and then added back increasing amounts of LDL. After cholesterol depletion, almost all of the SREBP2 was activated to its cleaved nuclear form (*Figure 6A*, top panel, lane 1). Incubation with increasing amounts of LDL for 3 hr suppressed SREBP2 activation, leading to a decrease in the

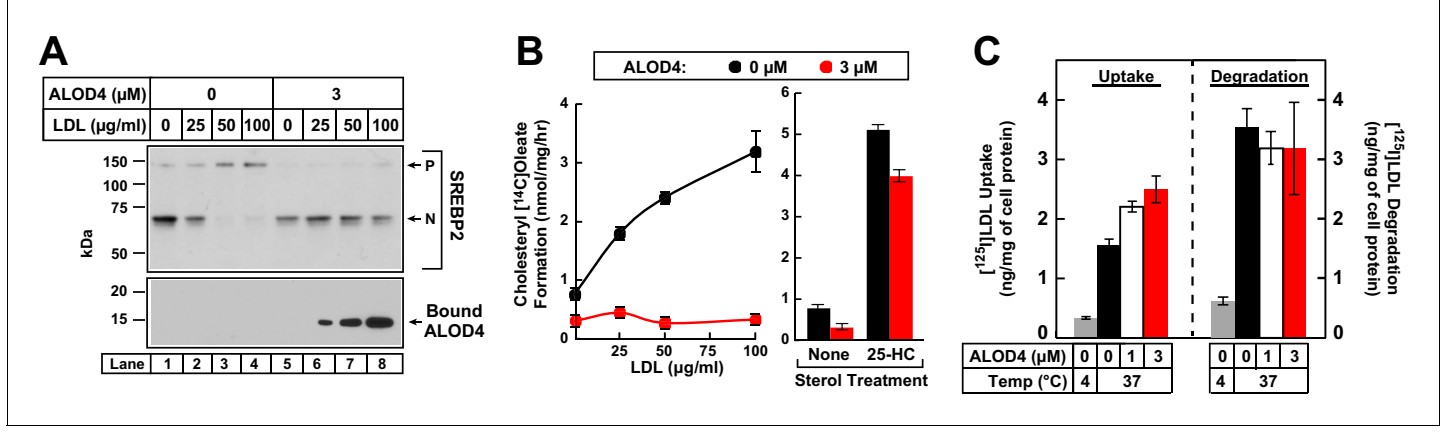

**Figure 6.** ALOD4 blocks PM-to-ER transport of LDL-derived cholesterol in CHO-K1 cells. (**A**) Immunoblot analysis of sterol-depleted CHO-K1 cells incubated with LDL in the absence or presence of ALOD4 proteins. On day 0, CHO-K1 cells were set up in medium B at a density of $3 \times 10^4$ cells/well of 48-well plates. On day 2, media was removed, and cells were washed twice with 500 µl PBS followed by the addition of 200 µl of medium E with 2% HPCD (sterol-depleting). After incubation for 1 hr at 37°C, media was removed, and sterol-depleted cells were washed twice with 500 µl of PBS followed by addition of 200 µl of medium E with the indicated concentration of human LDL in the absence or presence of 3 µM ALOD4. After incubation for 3 hr at 37°C, the cells were harvested and subjected to immunoblot analysis for SREBP2 and cell surface-bound ALOD4, as described in Materials and methods. (**B**) Cholesterol esterification in the presence of ALOD4. On day 0, CHO-K1 cells were set up in medium B at a density of $2.5 \times 10^5$ cells/60 mm dish. On day 2, media was removed, and cells were washed twice with 2 ml of PBS followed by addition of 2 ml of medium D. On day 3, media was removed, and cells were washed with 2 ml of PBS followed by addition of 1 ml of medium E with the indicated concentration of human LDL or 5 µg/ml of 25-HC (in ethanol), all in the absence or presence of 3 µM ALOD4. After incubation for 1 hr at 37°C, each monolayer was supplemented with 0.2 mM of sodium [14C]oleate (7759 dpm/nmol), and incubated for an additional 2 hr at 37°C. The cells were then harvested, and their levels of [14C]cholesteryl oleate and [14C]triglycerides were measured as described in Materials and methods. The levels of [14C]triglycerides formed at 0 and 100 µg/ml LDL treatment conditions were 38.6 and 39.3 nmol/mg/h, respectively, in the absence of ALOD4, and 31.2 and 30.7 nmol/mg/h, respectively, in the presence of 3 µM ALOD4. Each data point or column represents the mean of cholesterol esterification measurements from three independent experiments, and error bars show the standard error. (**C**) LDL uptake and degradation in the presence of ALOD4. CHO-K1 cells were set up on day 0 and treated on day 2 exactly as described in (**B**). On day 3, media was removed, and cells were washed with 2 ml of PBS followed by addition of 1 ml of medium E containing 50 µg/ml of [125I]LDL (35.3 cpm/ng) in the absence or presence of 1 or 3 µM of ALOD4. The cells were incubated for 3 hr at either 4°C or 37°C, after which LDL uptake and degradation was determined as described in Materials and methods. Each column represents the mean of measurements from three independent experiments, and error bars show the standard error. P = precursor form of SREBP2; N = cleaved nuclear form of SREBP2.

cleaved nuclear form and a corresponding increase in the uncleaved precursor form (*Figure 6A*, *top panel lanes 2–4*). However, addition of 3 μM ALOD4 during the 3 hr LDL incubation period completely blocked the suppression of SREBP2 activation, and only the cleaved nuclear form was detected (*Figure 6A*, *top panel lanes 5–8*). We also measured the levels of ALOD4 that had bound to the PMs of LDL-treated cells. After cholesterol depletion, no binding of ALOD4 was detected (*Figure 6A*, *bottom panel, lane 5*), indicating levels of accessible PM cholesterol were low. As increasing amounts of LDL were added, a steady increase in PM-bound ALOD4 was observed, indicating a rise in levels of accessible PM cholesterol (*Figure 6A*, *bottom panel lanes 6–8*). Although LDL-derived cholesterol can reach the PM, its subsequent transport to ER is blocked due to sequestration at the PM by ALOD4.

We were also able to show that ALOD4 blocks LDL-derived cholesterol from reaching the ER by using another measure of cholesterol delivery to ER, namely the activity of acyl-CoA acyltransferase (ACAT) (*Goldstein et al., 1983*). ACAT is an ER enzyme that esterifies some of the LDL-derived cholesterol that reaches the ER. CHO-K1 cells were first depleted of cholesterol by incubation in lipoprotein-poor LPDS along with compactin, an inhibitor of cholesterol biosynthesis (*Brown et al., 1978*). We then added increasing amounts of LDL together with [$^{14}$C]oleate, and after 3 hr, processed the cells for measurement of cholesteryl [$^{14}$C]oleate. As shown in *Figure 6B*, cholesteryl [$^{14}$C] oleate formation increased in a dose-dependent manner as increasing amounts of LDL were added (left panel, *black filled circles*), but was completely blocked when 3 μM ALOD4 was included during the incubation with LDL and [$^{14}$C]oleate (left panel, *red filled circles*). In contrast to the all-or-none effects observed with LDL, ALOD4 had little effect on 25-HC mediated cholesteryl [$^{14}$C]oleate formation (*Figure 6B*, *right panel*). This is likely because cholesterol-binding toxins like ALOD4 do not bind 25-HC (*Sokolov and Radhakrishnan, 2010*), and thus the oxysterol is free to enter cells and travel to the ER where it directly activates ACAT to esterify the cholesterol in that membrane (*Cheng et al., 1995*).

The increase in PM-bound ALOD4 detected after LDL addition (*Figure 6A*) argues against ALOD4 inhibiting the uptake or degradation of LDL; nevertheless the nature of ALOD4's interaction with cholesterol in membranes gave us some concerns. ALOD4 could potentially bind to regions of the PM containing clathrin-coated pits and block LDL internalization. ALOD4 could also bind to cholesterol in the lipid monolayer coating the LDL surface and then block LDL's binding to the LDL receptor or processing in lysosomes. To allay these concerns, we used previously described methods (*Goldstein and Brown, 1974*) to directly measure the uptake and degradation of LDL in the presence of ALOD4. CHO-K1 cells were depleted of cholesterol by incubation in lipoprotein-poor LPDS along with compactin. We then added 50 μg/ml [$^{125}$I]LDL, a concentration at which SREBP2 activation was completely suppressed in the absence of ALOD4 and not suppressed at all in the presence of 3 μM ALOD4 (*Figure 6A*, *upper panel,* compare *lane 3* to *lane 7*). After incubation for 3 hr with increasing concentrations of ALOD4, we processed the cells for measurement of surface-bound plus internalized [$^{125}$I]LDL (a measure of LDL uptake) and [$^{125}$I]monoiodotyrosine released into medium (a measure of degradation of LDL). As shown in *Figure 6C*, when incubated at 37°C, the uptake of [$^{125}$I]LDL increased as increasing concentrations of ALOD4 were added (*left panel*), a result that is consistent with increased expression of the LDL receptor induced by ALOD4 treatment (*Figure 3D*). The degradation of [$^{125}$I]LDL persisted even at the highest added concentration of 3 μM ALOD4 (*Figure 6C*, *right panel*). As a negative control, incubation of the cells at a temperature that blocks endocytosis (4°C) suppressed both [$^{125}$I]LDL uptake and degradation by more than 75% (*Figure 6C*, *gray bars*).

## Cholesterol trafficking in NPC1-deficient cells

We next sought to determine whether ALOD4 would affect PM-to-ER cholesterol transport in cells deficient in Niemann-Pick C1 (NPC1), a lysosomal membrane protein that is critical for transporting LDL-derived cholesterol out of lysosomes (*Liscum and Faust, 1987*; *Pentchev, 2004*). In NPC1-deficient cells, LDL-derived cholesterol does not reach the PM to fill up its pool of accessible cholesterol (*Das et al., 2013*). The PMs of NPC1-deficient cells must thus rely on biosynthesis to meet their cholesterol requirements. For this experiment, we used a previously described mutant CHO-K1 cell line (*Wojtanik and Liscum, 2003*) that had no detectable levels of NPC1 protein (*Figure 7A*, *top panel*). Consistent with previous studies, we also show here that LDL was unable to stimulate cholesterol esterification in these NPC1-deficient cells (*Figure 7A*, *black bars*). In contrast, stimulation of

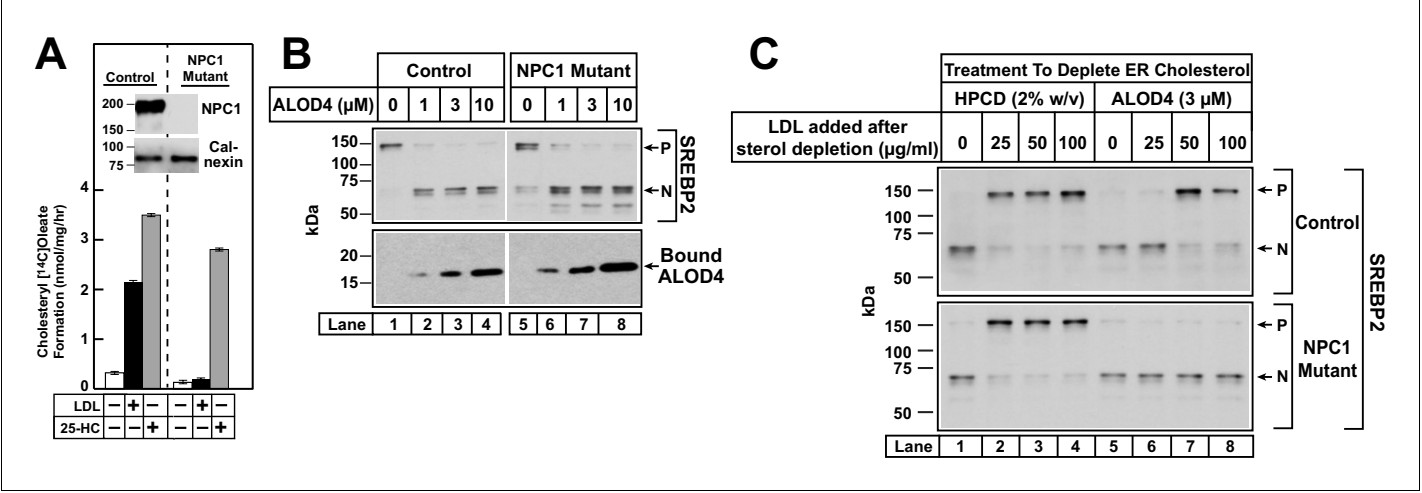

**Figure 7.** ALOD4 blocks PM-to-ER transport of cholesterol in NPC1-deficient CHO-K1 cells. (**A**) Cholesterol esterification in control and NPC1-deficient CHO-K1 cells. On day 0, cells were set up in medium B at a density of $2.5 \times 10^5$ cells/60 mm dish. On day 2, media was removed, and cells were washed twice with 2 ml of PBS followed by addition of 2 ml of medium D. On day 3, media was removed, and cells were washed with 2 ml of PBS followed by addition of 1 ml of medium E in the absence or presence of 50 µg/ml of human LDL or 4 µg/ml of 25-HC (in ethanol). After incubation for 2 hr at 37°C, each monolayer was supplemented with 0.2 mM of sodium [14C]oleate (3913 dpm/nmol), and incubated for an additional 2 hr. The cells were then harvested, and their levels of [14C]cholesteryl oleate and [14C]triglycerides were measured as described in Materials and methods. The levels of [14C]triglycerides formed at 0, 50 µg/ml LDL, and 4 µg/ml 25-HC treatment conditions were 21.8, 22.1 and 18.7 nmol/mg/h, respectively, for control CHO-K1 cells, and 24.7, 22.3, and 19.7 nmol/mg/h, respectively, for NPC1-deficient CHO-K1 cells. Each column represents the mean of cholesterol esterification measurements from three independent experiments, and error bars show the standard error. (*Inset*) On day 0, control and NPC1-deficient CHO-K1 cells were set up in medium B at a density of $3 \times 10^4$ cells/well of 48-well plates and $6 \times 10^4$ cells/well of 48-well plates, respectively. On day 2, media was removed, and cells were harvested and subjected to immunoblot analysis of the indicated proteins as described in the Materials and methods. (**B**) Immunoblot analysis of control and NPC1-deficient CHO-K1 cells after incubation with ALOD4. Cells were set up on day 0 in lipoprotein-rich FCS as described in A (*inset*). On day 2, media was removed, and cells were washed twice with 500 µl of PBS followed by addition of 200 µl of lipoprotein-rich medium C with the indicated concentrations of ALOD4. After incubation for 1 hr at 37°C, the cells were harvested and subjected to immunoblot analysis as described in the Materials and methods. (**C**) Suppression of SREBP2 activation in control and NPC1-deficient CHO-K1 cells after induction with HPCD or ALOD4. Cells were set up on day 0 as described in A (*inset*). On day 2, media was removed, and cells were washed twice with 500 µl of PBS followed by addition of 200 µl of medium E with 2% HPCD or 3 µM ALOD4. After incubation for 1 hr at 37°C, media was removed, and cells were washed twice with 500 µl of PBS, followed by addition of 200 µl of medium C with the indicated amount of LDL, in the absence or presence of 3 µM ALOD4. After incubation for 3 hr at 37°C, the cells were harvested and subjected to immunoblot analysis as described in the Materials and methods. *P* = precursor form of SREBP2; *N* = cleaved nuclear form of SREBP2.

cholesterol esterification by 25-HC occurred in both control and NPC1-deficient cells (*Figure 7A*, *gray bars*). We then set up control and NPC1-deficient cells in lipoprotein-rich FCS and incubated the cells with ALOD4 for 1 hr. We observed similar binding of ALOD4 to their PMs (*Figure 7B*, *bottom panel, compare lanes 1–4 to lanes 5–8*) and identical concentration dependences for triggering SREBP2 activation (*Figure 7B*, *top panel, compare lanes 1–4 to lanes 5–8*). This result suggests that the pool of PM cholesterol that is transported to ER to suppress SREBP2 activation is of a similar magnitude in control and NPC1-deficient cells.

By blocking PM-to-ER cholesterol transport, ALOD4 provides a new way to rapidly deplete ER cholesterol without significantly altering total cellular cholesterol levels. In contrast, the commonly used method of HPCD treatment to rapidly deplete ER cholesterol also lowers total cellular cholesterol by 30–40% (*Figures 3C* and *5*). Moreover, the use of HPCD to study cholesterol regulation in NPC1-deficient cells is confounded by HPCD's ability to rescue the lysosomal cholesterol accumulation caused by NPC1-deficiency by as yet undefined mechanisms (*Abi-Mosleh et al., 2009*; *Liu et al., 2009*). We thus decided to compare the effects of these two methods of sterol depletion in NPC1-deficient cells. As shown in *Figure 7C*, sterol depletion by HPCD triggered the activation of SREBP2 in both control CHO-K1 cells (*upper panel, lane 1*) and NPC1-deficient cells (*lower panel, lane 1*). Addition of LDL suppressed the activation of SREBP2 in control CHO-K1 cells (*upper panel, lanes 1–4*). LDL treatment also suppressed the activation of SREBP2 in NPC1-deficient cells (*lower*

*panel, lanes 1–4*). This paradoxical result is consistent with an earlier study where HPCD treatment overcame the inability of NPC1-deficient cells to transport LDL-derived cholesterol out of lysosomes to ER (*Abi-Mosleh et al., 2009*). A different result was obtained when ALOD4 was used to deplete ER cholesterol from NPC1-deficient cells. Consistent with the results of *Figure 7B*, incubation with 3 μM ALOD4 for 1 hr triggered the activation of SREBP2 in both control cells (*Figure 7C*, *upper panel, lane 5*) and NPC1-deficient cells (*Figure 7C*, *lower panel, lane 5*). We then replaced the ALOD4-containing medium with medium containing increasing amounts of LDL. Rapid dissociation of ALOD4 from PMs (*Figure 4C–D*) allowed LDL-derived cholesterol to transit through the PM and reach the ER to suppress the activation of SREBP2 in control cells (*Figure 7C*, *upper panel, lanes 5–8*), but not in NPC1-deficient cells (*Figure 7C*, *lower panel, lanes 5–8*). Thus, unlike HPCD, ALOD4 does not overcome the post-lysosomal cholesterol trafficking defect caused by NPC1-deficiency.

## Cholesterol delivered to PM does not reach ER in presence of ALOD4

We next examined the effect of ALOD4 when cholesterol was added to cells in complexes with methyl-β-cyclodextrin (MCD), which likely delivers cholesterol directly to PM without passing through lysosomes (*Abi-Mosleh et al., 2009*; *Das et al., 2014*). We first depleted cells of cholesterol by incubation with HPCD, triggering activation of SREBP2 (*Figure 8A*, *top and middle panels, lanes 1 and 5*). This activation of SREBP2 was suppressed when we added cholesterol/MCD complexes (*Figure 8A*, *top panel, lanes 1–4*), but suppression was blocked in the presence of 3 μM ALOD4 (*Figure 8A*, *middle panel, lanes 1–4*). After cholesterol depletion, no binding of ALOD4 was detected (*Figure 8A*, *bottom panel lane 1*), indicating levels of accessible PM cholesterol were low. As increasing amounts of cholesterol/MCD were added, we observed an increase in PM-bound ALOD4 (*Figure 8A*, *bottom panel lanes 1–4*), indicating that levels of accessible PM cholesterol had

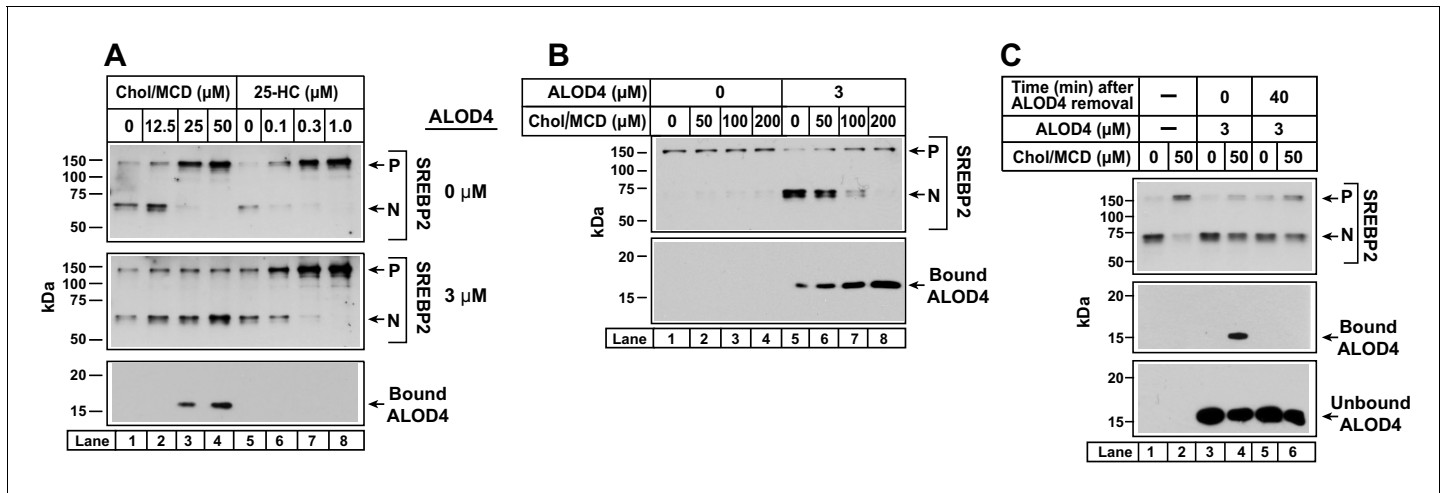

**Figure 8.** ALOD4 blocks PM-to-ER transport of cholesterol delivered directly to PM. (**A**) Immunoblot analysis of sterol-depleted CHO-K1 cells incubated with cholesterol/MCD complexes in the absence or presence of ALOD4. On day 0, CHO-K1 cells were set up in medium B at a density of $6 \times 10^4$ cells/well of 48-well plates in medium B. On day 1, media was removed, and cells were washed twice with 500 μl of PBS followed by addition of 200 μl of medium E with 2% HPCD (sterol-depleting). After incubation for 1 hr at 37°C, media was removed, and cells were washed twice with 500 μl of PBS followed by addition of 200 μl of medium E with the indicated amounts of cholesterol/MCD or 25-HC/ethanol, in the absence or presence of 3 μM ALOD4. (**B**) Triggering of SREBP2 activation by ALOD4 is overcome by addition of excess cholesterol/MCD. On day 0, CHO-K1 cells were set up in medium B at a density of $6 \times 10^4$ cells/well of 48-well plates. On day 1, media was removed, and cells were washed with 500 μl of PBS followed by addition of 200 μl of medium C with the indicated concentration of cholesterol/MCD in the absence or presence of 3 μM ALOD4. (**A–B**) After incubation for 4 hr at 37°C (**A**) or 1 hr at 37°C (**B**), the cells were harvested and subjected to immunoblot analysis of SREBP2 and cell surface-bound ALOD4 as described in Materials and methods. (**C**) ALOD4 is not internalized by sterol-depleted cells. CHO-K1 cells were set up on day 0 and depleted of sterols on day 1 exactly as described in (**A**). After sterol depletion, media was removed, and cells were washed twice with 500 μl of PBS followed by addition of 200 μl of medium E without or with 50 μM cholesterol/MCD, in the absence or presence of 3 μM ALOD4. After incubation for 4 hr at 37°C, media was collected, cells were washed twice with 500 μl of PBS, and either harvested immediately or incubated with 200 μl of medium C without ALOD4 for 40 min at 37°C, and then harvested. Equal aliquots of cells and media (10% of total) were subjected to immunoblot analysis as described in Materials and methods. *P* = precursor form of SREBP2; *N* = cleaved nuclear form of SREBP2.

increased. This suggests that cholesterol/MCD complexes deliver their cholesterol to the PM, where it is immediately sequestered by ALOD4, preventing its movement to the ER to suppress SREBP activation. In contrast, ALOD4 had no effect on the ability of 25-HC to suppress SREBP2 activation (*Figure 8A*, *top and middle panels, lanes 5–8*), and we detected no binding of ALOD4 to PMs after addition of 25-HC (*bottom panel, lanes 5–8*). This result is consistent with earlier studies showing that (i) cholesterol-binding toxins like ALOD4 bind cholesterol, but not oxysterols like 25-HC (*Sokolov and Radhakrishnan, 2010*); (ii) addition of 25-HC does not increase cholesterol in the whole cell or in ER membranes (*Radhakrishnan et al., 2008*), and (iii) 25-HC suppresses SREBP2 transport to Golgi by binding not to the cholesterol sensor Scap, but to Insigs, ER retention proteins (*Radhakrishnan et al., 2007*).

In *Figure 8B*, we tested whether ALOD4's ability to sequester cholesterol delivered to PM in MCD complexes could be saturated. We incubated CHO-K1 cells that had been growing in FCS with increasing amounts of cholesterol/MCD in the absence or presence of 3 µM ALOD4 for 1 hr, and then processed the cells for immunoblot analysis. In the absence of ALOD4, there was no triggering of SREBP2 activation since cellular sterol levels were high (*top panel, lanes 1–4*). When ALOD4 was added, it bound to PMs (*bottom panel, lane 5*) and prevented the PM cholesterol from traveling to the ER, thereby triggering SREBP2 activation (*top panel, lane 5*). When further supplemented with 50 µM cholesterol/MCD, the cholesterol levels in PM increased as judged by an increase in bound ALOD4 (*bottom panel, lane 6*). However, SREBP2 activation was not suppressed since all of the added cholesterol was likely sequestered by ALOD4 at PM and prevented from traveling to ER (*top panel, lane 6*). Supplementation with 100 µM and 200 µM cholesterol/MCD delivered even more cholesterol to PM, as judged by further increases in bound ALOD4 (*bottom panel, lanes 7 and 8*). In these cases, the additional cholesterol in PM was in excess of ALOD4's sequestering capacity, leading to transport of cholesterol to ER and suppression of SREBP2 activation (*top panel, lanes 7 and 8*).

In a final experiment, we examined whether ALOD4 added to the extracellular medium of CHO-K1 cells was internalized during the treatment conditions of the experiments in the current study. We have already shown that when added to sterol replete cells for 1 hr, ALOD4 bound to PM (*Figure 4D*, *lower panel lane 2*) but the PM-bound ALOD4 rapidly dissociated, with no detectable signal after 30 min (*Figure 4D–E*). If ALOD4 had been internalized during the 1 hr incubation period, the immunoblot analysis and fluorescence assays would have revealed residual bound ALOD4 (*Figure 4D–E*). We next measured whether ALOD4 could be internalized by sterol depleted cells (*Figure 8C*). We first depleted cells of cholesterol by incubation with HPCD for 1 hr, after which we incubated the cells without or with 50 µM cholesterol/MCD complexes for 3 hr. As expected, SREBP2 activation was triggered by sterol depletion, and this activation was suppressed by addition of cholesterol (*top panel, lanes 1 and 2*). When we included 3 µM ALOD4 during the 3 hr incubation period (*bottom panel, lanes 3–6*), the added cholesterol was trapped at the PM by the binding of ALOD4 (*middle panel, lane 4*) and was unable to reach the ER to suppress SREBP2 activation (*top panel, lane 4*). For a parallel set of cells, we removed the ALOD4-containing medium after the 3 hr incubation, and replaced it with medium without ALOD4. After 40 min, a time period that is sufficient for complete dissociation of PM-bound ALOD4 (*Figure 4*), cells were harvested and subjected to immunoblot analysis. We observed no bound ALOD4 in the condition where cholesterol was added (*middle panel, lane 6*), suggesting that all of the PM-bound ALOD4 had dissociated during the 40 min period and that no detectable level of ALOD4 had been internalized during the 3 hr incubation period. Restoration of PM-to-ER cholesterol transport after ALOD4 dissociation was not sufficient to suppress the activation of SREBP2 in the short 40 min duration of this experiment, a result that is consistent with our previous observation in *Figure 3D*.

## Discussion

Our current understanding of intracellular cholesterol trafficking has been largely made possible by tools that inhibit specific steps of this pathway (see *Figure 9*). These tools have been discovered through the study of genetic disorders or small molecule screens. Cells from individuals with familial hypercholesterolemia revealed how LDL receptors enable the uptake of cholesterol-rich LDL by a process called receptor-mediated endocytosis (*Brown and Goldstein, 1986*). Cells from individuals with Wolman disease or Cholesteryl Ester Storage disease revealed how lysosomal acid lipase converts cholesteryl esters liberated from the endocytosed LDL to unesterified cholesterol, a form that

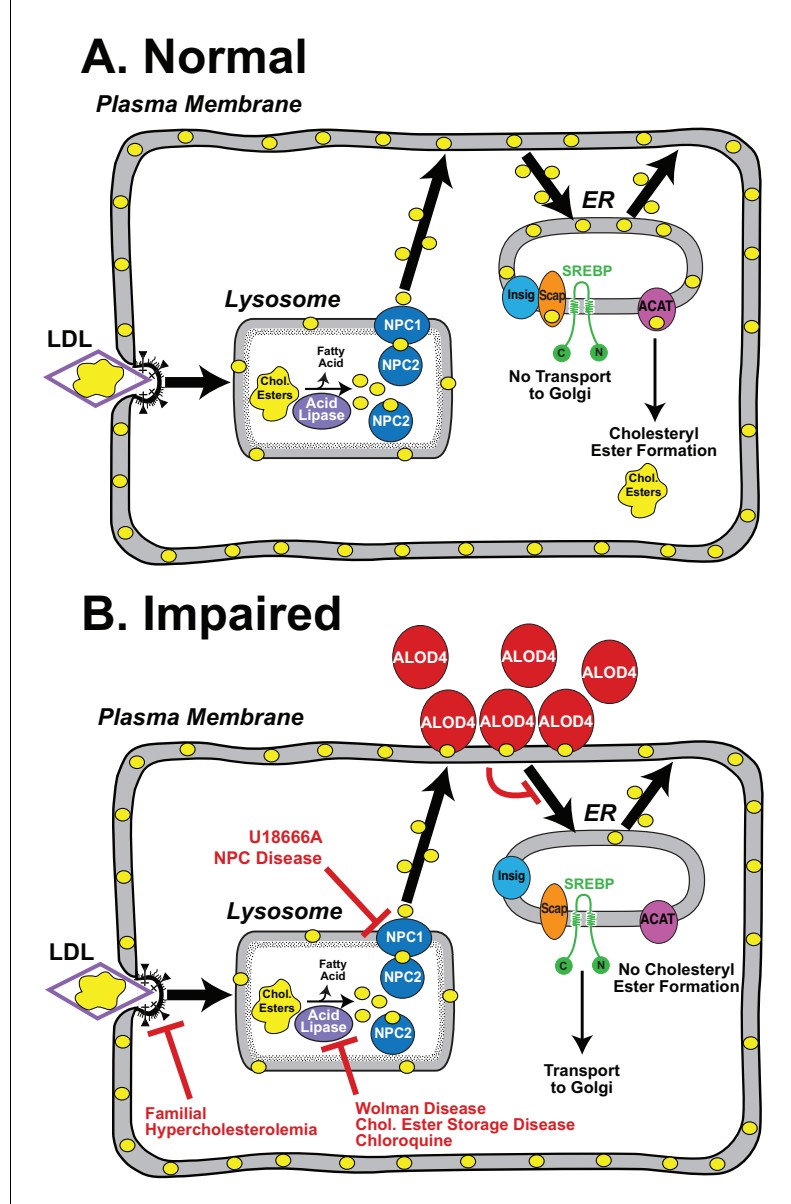

**Figure 9.** Model for the intracellular itinerary of LDL-derived cholesterol. (**A**) Normal trafficking. LDL particles containing cholesteryl esters are internalized by sterol-depleted cells through receptor-mediated endocytosis. The internalized LDL is degraded in lysosomes, and its cholesteryl esters (chol. esters) are hydrolyzed by acid lipase to generate unesterified cholesterol (*yellow circles*) and fatty acids. Two lysosomal proteins, NPC1 and NPC2, mediate the exit of LDL-derived cholesterol from lysosomes. Cholesterol is transported first to the sterol-depleted PM to replenish its cholesterol until optimal levels are reached. Excess PM cholesterol then traffics to regions of the ER where the SREBP regulatory network is located and where cholesterol levels are below a threshold concentration. After expanding ER cholesterol past the threshold level, excess cholesterol binds to SCAP, the ER cholesterol sensor, that is in complexes with SREBPs. Binding to cholesterol changes the conformation of Scap promoting its interaction with Insigs, ER retention proteins, and preventing recruitment of the Scap-SREBP complex into CopII vesicles for transport to Golgi. Proteolytic activation of SREBPs in Golgi does not occur, and genes encoding enzymes for cholesterol synthesis and uptake are shut down, thus maintaining cholesterol homeostasis. Excess ER cholesterol can be transported to the PM or can be converted by ACAT, an ER enzyme, to cholesteryl esters and stored in intracellular lipid droplets. (**B**) Genetic disorders or engineered tools that impair trafficking of LDL-cholesterol at various steps in its itinerary. See *Discussion* for a description of each of the impairments.

can be used by cells (*Brown et al., 1976*; *Goldstein et al., 1975*; *Sloan and Fredrickson, 1972*; *Wolman et al., 1961*). The activity of lysosomal acid lipase, and other lysosomal degradative processes, are also blocked by chloroquine (*Goldstein et al., 1975*). Cells from individuals with Niemann-Pick C disease revealed how two lysosomal proteins, NPC1 and NPC2, move the unesterified cholesterol generated by lysosomal acid lipase out of lysosomes (*Carstea et al., 1997*; *Kwon et al., 2009*; *Liscum and Faust, 1987*; *Naureckiene et al., 2000*). This last step is also blocked by U18666A, a cationic amphiphile that binds to NPC1 and inhibits cholesterol transport out of lysosomes (*Cenedella, 2009*; *Liscum and Faust, 1989*; *Lu et al., 2015*). Despite these advances, little is known about how cholesterol moves to other organelles after it exits the lysosome. The current study introduces ALOD4 as a new inhibitor to understand steps in the post-lysosomal transport of cholesterol.

ALOD4 is a non-lytic peptide derived from a bacterial toxin that binds to accessible cholesterol in membranes (*Gay et al., 2015*). Other versions of these toxins have been used in end-point assays to measure changes in accessible cholesterol in PM and ER membranes purified from cultured cells (*Das et al., 2014*; *Sokolov and Radhakrishnan, 2010*). These earlier studies showed that the distribution of cholesterol between inaccessible and accessible pools in PM and ER membranes plays crucial roles in controlling cellular cholesterol levels. In the current study with live cells, we found that ALOD4 does much more than just serve as a passive reporter of accessibility of PM cholesterol. When added in the extracellular medium of cells that were replete with cholesterol, ALOD4 rapidly bound PM cholesterol and prevented its movement to ER. As a result, ER cholesterol levels dropped below a threshold concentration of ~5 mole% cholesterol, thereby triggering activation of SREBP transcription factors (*Figures 2–5*). ALOD4 did not bind to PMs of cells that were depleted of cholesterol (*Figures 6A* and *8*). When we attempted to replenish sterol-depleted cells with MCD/cholesterol complexes that deliver cholesterol directly to PMs, ALOD4 immediately bound this newly delivered cholesterol in PMs, and prevented its movement to ER to suppress SREBP2 activation (*Figure 8*). A potentially useful feature of ALOD4 is that by sequestering PM cholesterol without extracting it from the membrane, ER cholesterol is depleted even though PM cholesterol levels and overall cellular cholesterol levels are not altered (*Figures 2D* and *5*). This sequestration is reversible, as removing ALOD4 from the medium results in rapid dissociation of PM-bound ALOD4 (*Figure 4D–E*), and restoration of PM-to-ER cholesterol transport (*Figure 7C*). In contrast, other cholesterol modulators like cyclodextrins, statins, or lipoprotein-poor serum, all irreversibly deplete cholesterol from both PM and ER membranes (*Das et al., 2014*, *2013*; *Radhakrishnan et al., 2008*).

ALOD4's specific inhibition of PM-to-ER cholesterol transport allowed us to clarify the trafficking route of LDL-derived cholesterol after it exits lysosomes. The post-lysosomal fate of LDL-derived cholesterol has been a subject of debate with some studies suggesting that it moves from lysosomes to PM (*Das et al., 2014*; *Lange and Steck, 1997*; *Xu and Tabas, 1991*), and other studies suggesting that it moves from lysosomes to ER (*Neufeld et al., 1996*; *Underwood et al., 1998*). In the presence of ALOD4, sterol-depleted cells internalized LDL and degraded it in lysosomes (*Figure 6C*), and the LDL-derived cholesterol exited the lysosomes and reached the PM (*Figure 6A*). At the PM, ALOD4 sequestered the LDL-derived cholesterol, preventing it from traveling to the ER and suppressing SREBP2 activation (*Figure 6A*), or stimulating the activity of ACAT (*Figure 6B*). These data support the model that LDL-derived cholesterol moves first from lysosomes to PM (*Figure 9*). Only after PM's cholesterol requirements are met does excess cholesterol travel from PM to ER. LDL-derived cholesterol from lysosomes cannot bypass the PM and travel directly to ER to shut down lipogenic genes, thus ensuring that the supply of cholesterol is not prematurely halted before the PM's needs are satisfied. Our simplified model shows LDL-derived cholesterol traveling directly from lysosomes to PM (*Figure 9*), but we cannot rule out the possibility that this cholesterol makes intermediate stops at other organelles such as peroxisomes on its way to PM (*Chu et al., 2015*), as long as these stops do not include the region of the ER housing the SREBP regulatory network or the ACAT enzyme. The PM to ER cholesterol transport route may also involve stops at other organelles. It should be noted that LDL-derived cholesterol is most likely delivered to the cytoplasmic leaflet of the PM, but must then flip to the outer leaflet of the PM to be accessible for binding to extracellularly added ALOD4. This step is likely not rate-limiting, since previous studies have suggested that equilibration of cholesterol between the bilayer leaflets of red cell membranes is rapid ($t_{1/2}$ of ~1 s) (*Steck et al., 2002*).

The kinetic studies of *Figure 4* allowed us to estimate the number of cholesterol molecules that must be sequestered in PM to trigger SREBP transport from ER to Golgi and proteolytic activation. At saturation, ~150 ng (~10 pmol) of ALOD4 bound to ~120,000 cells in a single well of a 48-well plate (*Figure 4C*). If we assume that each molecule of ALOD4 sequesters one molecule of cholesterol, we calculate that ALOD4 binds ~0.08 fmol of cholesterol/cell. In an earlier study, we estimated that a CHO-K1 cell growing in lipoprotein-rich FCS contains ~10 fmol of cholesterol (*Radhakrishnan et al., 2008*). The LIPID MAPS consortium quantified the lipidome of a mouse macrophage and reported that a macrophage cell contains 1000–2000 pmol cholesterol/μg DNA (*Dennis et al., 2010*). Using this value and previous measurements of ~10 picograms DNA/mouse cell (*Collins, 1978*; *Collins et al., 1980*), we calculate that a mouse macrophage contains ~10–20 fmol of cholesterol, similar to the estimated value for CHO-K1 cells. While being mindful of the many assumptions made in estimating these values, our results suggest that sequestering just a small fraction (~1%) of cellular cholesterol at the PM and preventing its movement to ER, can trigger activation of SREBPs in an all-or-none manner. Treatment with HPCD, a commonly used tool to study cholesterol regulation, lowers cellular cholesterol by 30–40%, and also triggers activation of SREBPs (*Figures 3C* and *5*). The results obtained with ALOD4 indicate that the cholesterol homeostatic machinery is sensitive to much more subtle changes in a small, labile fraction of PM cholesterol. It remains to be determined whether this small fraction represents a critical sub-pool of the previously determined accessible PM cholesterol pool (~15 mole% of PM lipids, *Das et al., 2014*). This metabolically active fraction of PM cholesterol can be maintained even in NPC1-deficient cells, which lack the ability to use LDL as a cholesterol source and must rely on cholesterol biosynthesis to stock its PM with optimal cholesterol levels. As shown in *Figure 7B*, ALOD4 triggered activation of SREBP2 in NPC1-deficient cells with identical concentration dependence as it did in control cells.

In cells replete with cholesterol, robust PM-to-ER cholesterol transport ensures that the ER cholesterol concentration is above a threshold value of 5 mole% of ER lipids (*Radhakrishnan et al., 2008*). At these high concentrations, cholesterol binds to the ER cholesterol sensor Scap, leading to formation of a complex with ER-resident Insigs, thus preventing Scap's recruitment into CopII coated vesicles and transport of SREBPs to Golgi where they can be proteolytically activated (*Figure 9A*). Incubating these cholesterol-replete cells with ALOD4 for 1 hr in the presence of compactin, an inhibitor of cholesterol synthesis, results in ER cholesterol dropping below the threshold concentration of 5 mole% (*Figure 5*). This drop in cholesterol concentration leads to cholesterol's dissociation from Scap, disassembly of the Scap-Insig complex, and promotes Scap's recruitment into CopII coated vesicles to transport and activate SREBPs (*Figure 9B*). How does ALOD4 cause a drop in ER cholesterol levels without altering overall cellular cholesterol? In the absence of synthesis, ER cholesterol is maintained at an equilibrium level by balancing the influx of cholesterol from the PM with the efflux of cholesterol out of ER to other organelles, including to the PM (*Lange et al., 2004*, *Das et al., 2014*). ALOD4's inhibition of the influx pathway, but not of the efflux pathway, disrupts this equilibrium and leads to ER cholesterol dropping below the threshold concentration of 5 mole%, even though cholesterol has not been removed from the PM (*Figure 5*). This suggests that when cells are replete with cholesterol, the ER continuously receives cholesterol from PM to offset losses of cholesterol due to efflux out of ER. This allows the ER to constantly sample the cholesterol content of PMs, and make adjustments via transcriptional regulation of cholesterol uptake and synthesis to ensure optimal PM cholesterol levels. How cholesterol moves from PM to ER is currently not understood, although PM proteins such as ABCA1, and vesicular and non-vesicular trafficking pathways have been implicated (*Holthuis and Menon, 2014*; *Lahiri et al., 2015*; *Mesmin and Maxfield, 2009*; *Yamauchi et al., 2015*).

In the experiments where we clarified the trafficking route of LDL-derived cholesterol, we purposefully included compactin to eliminate biosynthesis as a source for PM cholesterol. This allowed us to determine that LDL-derived cholesterol, once released from lysosomes, is directed first to the PM to ensure optimal cholesterol levels in that membrane before expanding the ER regulatory pool to downregulate cholesterol synthesis and uptake. A similar strategy is likely employed for newly synthesized cholesterol in ER, which must also be first directed to PM to meet the needs of that membrane before expanding the ER regulatory pool to shut down cholesterol synthesis and uptake. How proteins involved in cholesterol synthesis, regulation, and esterification are spatially organized in the ER, and how cholesterol is moved from ER to PM is currently not known. In addition to the

uses outlined in the current study, ALOD4 could also help to reveal the transport pathways that originate from the ER and deliver cholesterol to PM.

## Materials and methods

### Materials

We obtained [$^{14}$C]oleic acid (59 mCi/mmol) and [$^3$H]cholesteryl oleate (30–60 Ci/mmol) from American Radiolabeled Chemicals, St. Louis, MO; Coomassie Brilliant Blue R-450 from BioRad, Inc., Hercules, CA; methyl-$\beta$-cyclodextrin (randomly methylated) (MCD) and hydroxypropyl-$\beta$-cyclodextrin (HPCD) from Cyclodextrin Technologies Development, Inc., Gainesville, FL; [$^{125}$I]NaI (17 Ci/mg) from PerkinElmer, Waltham, MA; protease inhibitor cocktail tablets (cOmplete, EDTA-free) from Roche, Indianapolis, IN; 25-hydroxycholesterol (25-HC) from Steraloids, Inc., Newport, RI; bovine serum albumin (BSA), cholesterol, fetal bovine serum (FCS), phenylmethanesulfonyl fluoride (PMSF), and tris (2-carboxyethyl) phosphine (TCEP) from Sigma, St. Louis, MO; and Alexa Fluor 488 C$_5$-maleimide and Alexa Fluor 647 C$_2$-maleimide from Thermo Fisher, Waltham, MA. Human low-density lipoprotein (LDL, density <1.019 g/mL (fixed rotor) or density <1.063 g/mL (zonal)) and newborn calf lipoprotein-deficient serum (LPDS, density >1.3 g/mL) were prepared by ultracentrifugation as described previously (*Goldstein et al., 1983*). Solutions of compactin and sodium mevalonate were prepared as described previously (*Brown et al., 1978*). Solutions of cholesterol/MCD complexes were prepared at a final cholesterol concentration of 2.5 mM and a cholesterol:MCD ratio of 1:10 as described previously (*Brown et al., 2002*). We obtained monoclonal anti-His antibody from Millipore, Billerica, MA, monoclonal anti-lactate dehydrogenase (LDH) from Epitomics, Burlingame, CA, polyclonal anti-E1 and monoclonal anti-NPC1 from Abcam, Cambridge, MA, and polyclonal anti-calnexin from Novus Biologicals, Littleton, CO. Monoclonal antibody IgG-7D4 against hamster SREBP-2 (amino acids 32–350) (*Yang et al., 1995*), monoclonal antibody IgG-20B12 against hamster SREBP-1 (amino acids 32–250) (*Rong et al., 2017*), and monoclonal antibody IgG-4H4 against hamster Scap (amino acids 1–767) (*Ikeda et al., 2009*) are described in the indicated references.

### Buffers and culture media

Buffer A contained 50 mM Tris-HCl (pH 7.5) and 1 mM TCEP. Buffer B is buffer A supplemented with 150 mM NaCl. Buffer C contains 10 mM Tris-HCl (pH 6.8), 100 mM NaCl, 1% (w/v) SDS, 1 mM EDTA, 1 mM EGTA, 20 µg/ml PMSF, and protease inhibitors (one tablet/20 ml). Solution A contains 150 mM NaCl and 0.3 mM EDTA. Medium A is a 1:1 mixture of Ham's F-12 and Dulbecco's modified Eagle's medium. Medium B is medium A supplemented with 100 units/ml penicillin, 100 µg/ml of streptomycin sulfate and 5% (vol/vol) FCS. Medium C is medium A supplemented with 5% (vol/vol) FCS. Medium D is medium A supplemented with 100 units/ml penicillin, 100 µg/ml of streptomycin sulfate, 50 µM compactin, 50 µM sodium mevalonate, and 5% (vol/vol) LPDS. Medium E is medium A supplemented with 50 µM compactin, 50 µM sodium mevalonate, and 5% (vol/vol) LPDS. Medium F is medium A supplemented with 5% (vol/vol) LPDS. Medium G is Dulbecco's modified Eagle's medium (low glucose) supplemented with 5% (vol/vol) FCS. Medium H is medium G supplemented with 100 units/ml penicillin and 100 µg/ml of streptomycin sulfate.

### Cell culture

Stock cultures of hamster CHO-K1 cells and 10–3 cells (mutant CHO-K1 cells that lack detectable mRNA for Niemann-Pick C1) (*Wojtanik and Liscum, 2003*) were maintained in monolayer culture at 37°C in 8.8% CO$_2$. Stock cultures of SV-589 cells were maintained in monolayer culture at 37°C in 5% CO$_2$. These environmental conditions were used for all experimental incubations, unless otherwise indicated. CHO-K1, SV-589, and 10–3 cells were obtained from American Type Culture Collection (CCL 61), Human Genetic Cell Repository (GM 639), and Laura Liscum (Tufts University), respectively. Each cell line was propagated, aliquoted, and stored under liquid nitrogen. To guard against potential genomic instability, an aliquot of each cell line is passaged for only 4 weeks (SV-589 and NPC 10–3) or 8 weeks (CHO-K1) before a fresh batch of cells is thawed and propagated for experimental use. All the cell lines have been confirmed to be free of mycoplasma contamination using the MycoAlert Mycoplasma Detection Kit (Lonza, Allendale, NJ).

## Protein purification and labeling

The following recombinant expression plasmids have been previously described: pALOFL encoding His$_6$-tagged signal-peptide deficient ALO (amino acids 35–512); pALOD4 encoding His$_6$-tagged domain 4 (amino acids 404–512) of ALO with two point mutations (S404C and C472A); and pALOD4 (Mut) encoding ALOD4 with six additional point mutations (S404C, C472A, G501A, T502A, T503A, L504A, Y505A, and P506A) (*Gay et al., 2015*). Recombinant ALO proteins were overexpressed in *E. Coli* and purified by nickel chromatography as described previously (*Gay et al., 2015*). ALOFL was further purified by gel filtration chromatography in buffer B, as described previously (*Gay et al., 2015*), and stored at 4°C until use. ALOD4-rich elution fractions from nickel chromatography (containing 150 mM NaCl) were combined and concentrated to a final volume of 15 mL using a 10,000 molecular weight cut-off Amicon Ultra centrifugal filter (Millipore). We then added 135 mL of NaCl-free buffer A to lower the NaCl concentration to ~15 mM, and loaded the mixture on a 1 ml anion-exchange chromatography column (HiTrap Q, GE Healthcare, Pittsburgh, PA). After washing with 20 column volumes of buffer A containing 50 mM NaCl, bound ALOD4 was eluted with buffer A containing 500 mM NaCl into a single 2 ml fraction. After dilution with buffer A to lower the NaCl concentration to 150 mM, ALOD4 was further diluted in buffer B to reach a final ALOD4 protein concentration of 2 mg/ml. This material was either directly used in experiments or supplemented with 20% (v/v) glycerol, flash frozen in liquid nitrogen, and stored at −80°C for later use. In some cases, the lone engineered cysteine on ALOD4 (at amino acid 404) was labeled with Alexa Fluor maleimide dyes as described previously (*Gay et al., 2015*). Degree of labeling was greater than 0.5 in all cases. Protein concentrations were measured using a NanoDrop instrument (Thermo Fisher) or a bicinchoninic acid kit (Thermo Fisher).

## Assays for cholesterol esterification

The rate of incorporation of [$^{14}$C]oleate into cholesteryl [$^{14}$C]oleate and [$^{14}$C]triglycerides in cultured CHO-K1 cells was measured as described previously (*Goldstein et al., 1983*).

## Assays for uptake and degradation of LDL

Human LDL was iodinated using Pierce pre-coated iodination tubes (Thermo Fisher) according to the manufacturer's instructions. Each tube, in a final volume of 500 µl of solution A, contained human LDL (5 mg) and [$^{125}$I]NaI (2 mCi). After incubation for 15 min at room temperature, 2 ml of solution A was added to the tube, and the entire mixture was loaded onto a PD-10 column (GE Healthcare) that had been pre-equilibrated with solution A. Elution fractions containing [$^{125}$I]LDL were pooled and subjected to dialysis for 16 hr against 6 L of solution A to further eliminate unincorporated [$^{125}$I] NaI. The dialyzed [$^{125}$I]LDL had a specific activity of 70.6 cpm/ng and was stored at 4°C. The uptake and proteolytic degradation of [$^{125}$I]LDL by cultured CHO-K1 cells was measured using previously described methods (*Goldstein and Brown, 1974*).

## Immunoblot analysis

After indicated treatments, media was removed from each well of 48-well plates, wells were washed twice with 500 µl PBS, and 200 µl of buffer C was added to each well. The plate was then placed on a shaker at room temperature for 20 min, after which the lysed cells were collected, mixed with 5x loading buffer, heated at 95°C for 10 min, and subjected to either 10% or 15% SDS/PAGE. The electrophoresed proteins were transferred to nitrocellulose filters using the Bio-Rad Trans Blot Turbo system, and subjected to immunoblot staining with the following primary antibodies: IgG-7D4 (10 µg/ml), anti-His (1:1000 dilution), IgG-20B12 (2 µg/ml), IgG-4H4 (0.2 µg/ml), anti-LDH (1:1000 dilution), anti-E1 (1:1000 dilution), anti-NPC1 (1:1000 dilution), and anti-calnexin (1:2000 dilution). Bound antibodies were visualized by chemiluminescence (Super Signal Substrate; Thermo Fisher) by using a 1:5000 dilution of donkey anti-mouse IgG (Jackson ImmunoResearch, West Grove, PA) or a 1:2000 dilution of anti-rabbit IgG (GE Healthcare) conjugated to horseradish peroxidase. Filters were exposed to Phoenix Blue X-Ray Film (F-BX810; Phoenix Research Products, Pleasanton, CA) at room temperature for 1–300 s or scanned using an Odyssey FC Imager (Dual-Mode Imaging System; 2 min integration time) and analyzed using Image Studio ver. 5.0 (LI-COR, Lincoln, NE).

## Quantitative real-time PCR

After indicated treatments of 100 mm dishes, media was removed, and cells were washed twice with 5 ml of PBS. After addition of 1 ml PBS, cells were harvested by scraping and transferred to 1.7 ml centrifuge tube. An aliquot (100 µl) was saved for immunoblot analysis. The remainder (900 µl) was subjected to centrifugation at 2000 x g for 10 min, after which the pellets were resuspended in 600 µl of Buffer RLT (RNeasy Mini Kit, Qiagen, Germantown, MD). Total RNA was prepared using the RNeasy Mini Kit (Qiagen) using the manufacturer's directions, and subjected to real time PCR analysis. The primer sequences used for PCR were as follows: HMG CoA reductase (Fwd: AGATACTGGA-GAGTGCCGAGAAA; Rev: TTTGTAGGCTGGGATGTGCTT), LDL receptor (Fwd: AGACACA TGCGACAGGAATGAG; Rev: GACCCACTTGCTGGCGATA), and Actin (Fwd: GGCTCCCAGCACCA TGAA; Rev GCCACCGATCCACACAGAGT).

## Quantification of cellular cholesterol

After indicated treatments of 60 mm dishes, media was removed, and cells were washed twice with 2 ml PBS. After addition of 1 ml PBS, cells were harvested by scraping and transferred to 1.7 ml centrifuge tubes. An aliquot (50 µl) was centrifuged at 4000 rpm for 10 min, followed by resuspension of the pellet in 100 µl of buffer C, lysis using a 22-gauge needle, and vigorous disruption on a shaker for 30 min. After using 20 µl of the resuspended pellet for quantifying protein content using a bicin-choninic acid assay kit, the remainder of the resuspended pellet (80 µl) was mixed with 5x loading buffer, heated at 95°C for 10 min, and then subjected to SDS/PAGE and immunoblot analysis as described above. The remaining 950 µl of the original cell sample was used for cholesterol measurements. Lipids were extracted using a chloroform/methanol (1:1, v/v) mixture, the organic solvent was evaporated under a gentle stream of nitrogen, and unesterified cholesterol was measured using the Amplex Red Cholesterol Assay Kit (Thermo Fisher).

## Quantification of PM and ER cholesterol

PM and ER membranes were purified and their cholesterol content was quantified as described previously (*Das et al., 2013*; *Radhakrishnan et al., 2008*).

## Quantification of fALOD4 bound to CHO-K1 cells

After indicated treatments, media was removed from each well of 48-well plates, wells were washed twice with 500 µl PBS to eliminate unbound fALOD4, and 200 µl of buffer C was added to each well. The plate was then placed on a shaker at room temperature for 20 min, after which the lysed cells including surface-bound fALOD4 were collected, and transferred to a 96-well plate (Greiner Bio-One, Monroe, NC). Solubilization of bound fALOD4 in SDS-containing buffer C eliminated fluorescence quenching effects that occur after binding of fALOD4 to cholesterol-containing membranes (*Gay et al., 2015*). The 96-well plate was stored in a −20°C freezer for at least 2 hr to eliminate bubbles, after which bound fALOD4 fluorescence was measured using a Tecan microplate reader (fALOD4-488: excitation wavelength: 495 nm, emission wavelength: 517 nm; fALOD4-647: excitation wavelength: 651 nm, emission wavelength: 672 nm). By measuring the fluorescence (at identical gain settings) from wells containing known amounts of fALOD4 in buffer C, we were able to quantify the amount of bound fALOD4.

## Reproducibility of data

All results were confirmed in at least three independent experiments conducted on different days using different batches of cells and different batches of purified ALOD4. The only exception was the large-scale study described in *Figure 5* which was conducted in triplicate on one occasion.

## Acknowledgements

We thank Mike Brown and Joe Goldstein for their mentorship and valuable suggestions. We thank Sarah Bayless and Anthony Yuan for their help during the early phases of this work; Donna Frias and Daphne Rye for excellent technical assistance; Lucie Batte, Shomanike Head, Camille Harry, and Lisa Beatty for cell culture assistance; Linda Donnelly and Don Anderson for assistance with [125]I-LDL uptake and degradation assays; Jessica Proulx and Feiran Lu for assistance with assays for

cholesterol esterification by the ACAT enzyme; Shreya Endapally and Gurpreet Arora for assistance with protein purification; Jeff McDonald for assistance with cholesterol quantification, Jeff Cormier and Tong Guo for assistance with quantitative real-time PCR, and Jay Horton, Russell DeBose-Boyd, and David Russell for valuable suggestions.

## Additional information

### Funding

| Funder | Grant reference number | Author |
|---|---|---|
| National Institutes of Health | HL20948 | Rodney Elwood Infante<br>Arun Radhakrishnan |
| Welch Foundation | I-1793 | Arun Radhakrishnan |
| American Heart Association | 12SDG12040267 | Arun Radhakrishnan |
| National Institutes of Health | T32DK007745 | Rodney Elwood Infante |

The funders had no role in study design, data collection and interpretation, or the decision to submit the work for publication.

### Author contributions

REI, Conceptualization, Data curation, Formal analysis, Funding acquisition, Investigation, Methodology, Project administration; AR, Conceptualization, Data curation, Formal analysis, Funding acquisition, Investigation, Methodology, Writing—original draft, Project administration, Writing—review and editing

### Author ORCIDs

Rodney Elwood Infante, http://orcid.org/0000-0003-2605-822X
Arun Radhakrishnan, http://orcid.org/0000-0002-7266-7336

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
