## [Decision Letter]

Thank you for submitting your article "Continuous transport of a small fraction of plasma membrane cholesterol to ER regulates total cellular cholesterol" for consideration by *eLife*. Your article has been favorably evaluated by Randy Schekman (Senior Editor) and three reviewers, one of whom is a member of our Board of Reviewing Editors. The reviewers have opted to remain anonymous.

The reviewers have discussed the reviews with one another and the Reviewing Editor has drafted this decision to help you prepare a revised submission.

Summary:

The authors have made an important discovery in cellular cholesterol metabolism. They have shown that a modified version of a cholesterol-binding cytolysin, ALO-D4, incubated with CHO cells, locks the accessible pool of cholesterol on the plasma membrane and prevents it from moving the ER. The diminished transport of cholesterol to the ER results in endoproteolytic processing of SREBP and activation of the SREBP pathway.

Essential revisions:

1) The authors are experts in measuring ER cholesterol in CHO cells. They should show, by measuring ER cholesterol in at least one experimental setting, that ALO-D4 binding to the plasma membrane reduces cholesterol in the ER. Also, show that the processing of SREBP is accompanied by the expected gene-expression changes.

2) Provide information on reproducibility of data and more detail on statistical analyses of data.

3) Show that ALO-D4 binding activates SREBP processing in at least one other cell type, for example human fibroblasts.

4) Show a dose-response curves for the impact of ALO-D4 binding on SREBP processing.

5) Provide more context to the contention that cholesterol travels from lysosomes to the plasma membrane and qualify the contention that ALO-D4 serves as an inhibitor of plasma membrane to ER transport. Transport between these organelles would be better presented as an equilibrium. ALOD4 sequesters PM cholesterol and therefore shifts this equilibrium.

6) Provide improved microscopy or delete it.

7) Cite or show data that ALO-D4 is not internalized.

---

## [Author Response]

*Essential revisions:*

*1) The authors are experts in measuring ER cholesterol in CHO cells. They should show, by measuring ER cholesterol in at least one experimental setting, that ALO-D4 binding to the plasma membrane reduces cholesterol in the ER. Also, show that the processing of SREBP is accompanied by the expected gene-expression changes.*

We thank the reviewers for suggesting this important experiment. We have now directly measured the effect of ALOD4 treatment on ER cholesterol levels. The result of this experiment, where we also measured ALOD4’s effect on PM cholesterol levels, is shown in a new Figure 5. For this experiment, we set up three sets of CHO-K1 cells in lipoprotein-rich FCS. We then maintained one set of cells in FCS, and incubated the other two sets with either ALOD4 or cholesterol-depleting HPCD. After treatment for 1 h, aliquots of cells from each set were subjected to immunoblot analysis for SREBP2, whole cell cholesterol quantification, PM purification and quantification of PM cholesterol (Das et al., 2013), and ER purification and quantification of ER cholesterol (Radhakrishnan et al., 2008). Consistent with the results of Figure 3, treatment of cells with ALOD4 or HPCD both triggered similar activation of SREBP2 (Figure 5, lanes2and3), however the treatments had distinct effects on cellular cholesterol distribution. Compared to the FCS-treated set, ALOD4 treatment did not significantly alter cholesterol levels in the whole cell or in PM, whereas HPCD treatment lowered total cellular cholesterol from ~33 µg/mg protein to ~20 µg/mg protein and PM cholesterol from 41 mole% of total PM lipids to ~31 mole% of total PM lipids (Figure 5). A different relationship was found when we quantified ER cholesterol levels. The ER cholesterol content of FCS-treated cells was ~ 8 mole% of total ER lipids and this value declined by a similar degree to ~ 4 mole% of total ER lipids when treated with ALOD4 or HPCD (Figure 5).

In a separate experiment, we show that ALOD4-induced triggering of SREBP processing is accompanied by expected increases in expression levels of HMG-CoA reductase and LDL receptor, both of which are SREBP target genes. These results are shown in a new Figure 3.

Combined, these new sets of data bolster our interpretation of ALOD4’s effect in the Discussion. Binding of ALOD4 to PMs blocks PM-to-ER cholesterol transport, resulting in an ~ 50% reduction of ER cholesterol levels even though PM cholesterol and overall cellular cholesterol do not change. The drop in ER cholesterol triggers activation of SREBP transcription factors, increasing expression of target genes to increase cholesterol synthesis and uptake.

*2) Provide information on reproducibility of data and more detail on statistical analyses of data.*

We now include detailed statistical analyses of all bar graph data in the paper. Individual data points for all triplicate studies are shown along with the standard error. *P* values are provided whenever statements of significance are made.

All results were confirmed in at least three independent experiments conducted on different days using different batches of cells and different batches of purified ALOD4. The only exception was the large-scale study described in Figure 5 which was conducted in triplicate on one occasion. We have included this information in a section titled “Reproducibility of data” in the Materials and methods.

*3) Show that ALO-D4 binding activates SREBP processing in at least one other cell type, for example human fibroblasts.*

The results of a new experiment showing that ALOD4 triggers the activation of SREBP1 and SREBP2 in human fibroblasts are now included in Figure 2.

*4) Show a dose-response curves for the impact of ALO-D4 binding on SREBP processing.*

A detailed dose response curve showing that increased ALOD4 binding leads to increasing processing of SREBP is shown in Figure 2. Less detailed dose response curves showing a similar result are shown in Figure 3, Figure 4 and Figure 7.

*5) Provide more context to the contention that cholesterol travels from lysosomes to the plasma membrane and qualify the contention that ALO-D4 serves as an inhibitor of plasma membrane to ER transport. Transport between these organelles would be better presented as an equilibrium. ALOD4 sequesters PM cholesterol and therefore shifts this equilibrium.*

The direct measurement of a decrease in ER cholesterol levels after ALOD4 treatment (Figure 5) strengthens our contention that ALOD4 serves as an inhibitor of plasma membrane to ER transport, and helps to clarify our interpretation that the LDL studies in Figure 6 and Figure 7 suggest a lysosome-PM-ER route for LDL cholesterol. We have made changes to the Discussion to reflect these points. We have also modified Figure 9 and the Discussion to more clearly convey how ALOD4 affects the equilibrium transport of cholesterol between PM and ER.

*6) Provide improved microscopy or delete it.*

We have deleted the microscopy figure.

*7) Cite or show data that ALO-D4 is not internalized.*

Studies by other groups have shown that the non-lytic domain 4 of PFO or ALO is not internalized, however these studies were carried out at 4°C or room temperature. Although suggestive, these earlier studies did not address whether ALOD4 would be internalized during the 37°C incubations of the current study. We have cited the suggestive earlier studies and also describe two experiments which suggest that ALOD4 is not internalized at 37°C – 1) When added to sterol-replete cells for 1 h at 37°C, ALOD4 bound to PM. The PM-bound ALOD4 rapidly dissociated, with no detectable signal after 30 min (Figure 4). If ALOD4 had been internalized during the 1 h incubation period, the internalized ALOD4 would have been detected as residual bound ALOD4. 2) We also measured whether ALOD4 could be internalized by sterol depleted cells (Figure 8). We first depleted cells of cholesterol by incubation with HPCD for 1 h, after which we incubated the cells without or with 50 µM cholesterol/MCD complexes for 3 h. As expected, SREBP2 activation was triggered by sterol depletion, and this activation was suppressed by addition of cholesterol (top panel, lanes 1 and 2). When we included 3 µM ALOD4 during the 3 h incubation period (bottom panel, lanes 3-6), the added cholesterol was trapped at the PM by the binding of ALOD4 (middle panel, lane 4) and was unable to reach the ER to suppress SREBP2 activation (top panel, lane 4). For a parallel set of cells, we removed the ALOD4-containing medium after the 3 h incubation, and replaced it with medium without ALOD4. After 40 min, a time period that is sufficient for complete dissociation of PM-bound ALOD4 (Figure 4), cells were harvested and subjected to immunoblot analysis. We observed no bound ALOD4 in the condition where cholesterol was added (middle panel, lane 6), suggesting that all of the PM-bound ALOD4 had dissociated during the 40 min period and that no detectable level of ALOD4 had been internalized during the 3 h incubation period.